# Protein disulfide isomerase cleaves allosteric disulfides in histidine-rich glycoprotein to regulate thrombosis

Keyu Lv[1,17], Shuai Chen[1,2,17], Xulin Xu [1,3,4,17], Joyce Chiu [5], Haoqing J. Wang[6,7], Yunyun Han [8], Xiaodan Yang[8], Sheryl R. Bowley[9], Hao Wang[1], Zhaoming Tang[10], Ning Tang[11], Aizhen Yang[12], Shuofei Yang[13], Jinyu Wang[14], Si Jin[15], Yi Wu[12], Alvin H. Schmaier[16], Lining A. Ju [6,7], Philip J. Hogg [5] & Chao Fang [1,3,4] ✉

The essence of difference between hemostasis and thrombosis is that the clotting reaction is a highly fine-tuned process. Vascular protein disulfide isomerase (PDI) represents a critical mechanism regulating the functions of hemostatic proteins. Herein we show that histidine-rich glycoprotein (HRG) is a substrate of PDI. Reduction of HRG by PDI enhances the procoagulant and anticoagulant activities of HRG by neutralization of endothelial heparan sulfate (HS) and inhibition of factor XII (FXIIa) activity, respectively. Murine HRG deficiency ($Hrg^{-/-}$) leads to delayed onset but enhanced formation of thrombus compared to *WT*. However, in the combined FXII deficiency ($F12^{-/-}$) and HRG deficiency (by siRNA or $Hrg^{-/-}$), there is further thrombosis reduction compared to $F12^{-/-}$ alone, confirming HRG's procoagulant activity independent of FXIIa. Mutation of target disulfides of PDI leads to a gain-of-function mutant of HRG that promotes its activities during coagulation. Thus, PDI-HRG pathway fine-tunes thrombosis by promoting its rapid initiation via neutralization of HS and preventing excessive propagation via inhibition of FXIIa.

Most antithrombotic drugs are associated with bleeding risks, albeit to different extent[1]. The essence is that hemostatic response is a highly dynamic process that is fine-tuned by multiple positive and negative regulatory mechanisms. As part of the normal defense processes, the clotting system is rapidly initiated during hemostasis following vessel damage to seal the breach and sustain the integrity of a closed, high-pressure circulatory system[2]. Concomitant with activated platelets that are recruited to the injury site as a major component of a developing thrombus, blood coagulation, triggered by tissue factor (TF) of the extrinsic pathway, is activated to produce thrombin resulting in fibrin generation. The clot formation is further propagated by the intrinsic pathway through activation of factor XII (FXII) on negatively charged surface such as polyphosphate released by activated platelets[3–5] and perhaps other entities such as microvesicles, exosomes, DNA, and RNA, ultimately leading to thrombus formation. Under physiological conditions there exist multiple mechanisms to maintain the clotting system in a quiescent state. For instance, the initiation phase is suppressed by TF pathway inhibitor (TFPI)[6], while the propagation phase mediated by activated FXII (FXIIa) is inhibited by C1 inhibitor[7] and histidine-rich glycoprotein (HRG)[8]. Additionally, procoagulant responses also are suppressed by vascular components such as heparan sulfate (HS) and other glycosaminoglycans which inhibit coagulation by binding antithrombin on the endothelial surface[9–11] etc. The initiation of thrombosis under pathologic condition requires the rapid overwhelming of the regulatory anticoagulant mechanisms. However, the biology underlying these events remain largely unknown. Understanding the molecular mechanisms whereby thrombus formation is fine-tuned is important for the design of more precise therapeutic strategies.

The secretion of protein disulfide isomerase (PDI) following vessel injury has been recently recognized as critical initiation signals that induce thrombus formation[2,12]. PDI is the prototypic member of thiol

isomerase family responsible for the catalysis of disulfide bond rearrangement[13]. Originally identified in the endoplasmic reticulum (ER), PDI is secreted by activated vascular cells and proposed as a "master switch"[13] to modify allosteric disulfide bonds which control the functions of thrombosis-related proteins where they reside[12]. This has been recognized as an important mechanism that regulates platelet activation and fibrin generation[13–17]. In addition, a clinical trial in advanced cancer patients showed that targeting extracellular PDI by isoquercetin reduced coagulation markers without causing major bleeding events[18,19]. However, the precise pathways through which PDI modulates the kinetics of thrombosis remain unknown. We performed mechanism-based kinetic trapping using PDI variants which have the C-terminal cysteine in the catalytic CGHC motif replaced by Ala to form stable disulfide-linked complexes with substrates to allow their identification by mass spectrometry[20]. HRG was identified to be a substrate of PDI redox activity using platelet-rich plasma.

HRG exhibits two potential functions during coagulation: on one hand, HRG inhibits the intrinsic pathway of coagulation by binding to FXIIa[21,22]; on the other hand, HRG attenuates the anticoagulant activity of antithrombin by competing for binding to endothelial HS[11,23,24]. In this study, we determined the effects of PDI-mediated modification of disulfides on the functions of HRG and their influence on thrombus formation. Our results show that cleavage of allosteric disulfides by PDI enhances the binding of HRG to HS and FXIIa, which differentially regulates thrombosis: at the early phase, PDI promotes HRG binding to endothelium to displace antithrombin allowing for rapid initiation of coagulation; at the later stage, PDI cleavage of HRG disulfides enhances the inhibition of FXIIa to prevent excessive clot formation. Mutation of these target disulfide bonds led to a gain-of-function mutant of HRG which enabled its activities. Thus, PDI modulation of allosteric disulfides in HRG represents a novel regulatory mechanism that fine-tunes the kinetics of arterial thrombus formation. This study provides mechanistic insights for the clinical effects of PDI inhibitors which reduce thrombosis without promoting bleeding[18,19].

## Results

### Plasma HRG is a substrate for PDI

We identified plasma HRG as a redox substrate of extracellular PDI using mechanism-based kinetic trapping in combination with 2D electrophoresis and mass spectrometry[20]. Based on the mechanisms of PDI-catalyzed disulfide reduction, we prepared 5 different variants of recombinant PDI (Supplementary Fig. S1). To validate the disulfide reaction between HRG and PDI variants, we analyzed the PDI-substrate complexes isolated from platelet-rich plasma by immunoblotting. When probed with anti-FLAG, which recognizes recombinant PDI, the three trapping variants with one or both C-terminal Cys in the CGHC motifs replaced by Ala (CACC, CCCA and CACA) formed stable disulfide-linked complexes with their substrates. These complexes migrated at higher molecular weight on non-reducing gel (Fig. 1a, left panel), compared to the unbound monomeric PDI migrating at the bottom. When probed with an anti-HRG antibody, HRG was detected between the 117 kD and 171 kD markers in the samples incubated with PDI trapping variants (Fig. 1a, central panel). The molecular weight corresponded to stable covalent complexes between HRG and PDI at 1:1 stoichiometry. Simultaneous visualization with secondary antibodies with different fluorochrome conjugates showed the colocalization of HRG (Red) with the PDI-substrate complexes (Green) (Fig. 1a, right panel). In contrast to the trapping variants, neither wt-PDI (CCCC) nor inert-PDI (AAAA), where all catalytic Cys were replaced with Ala, reacted with HRG.

Endoplasmic reticulum protein 57 (ERp57) is a member in the thiol isomerase family which shares a similar domain structure as PDI with two catalytic CGHC motifs in domains $a$ and $a'$[13,15]. We determined if ERp57 also reacted with HRG using the same strategy. Immunoblotting showed that trapping variants of ERp57 (CACC, CCCA and CACA)

formed stable disulfide complexes with their substrates which migrated with a distinct pattern from that of the PDI trapping variants (Fig. 1b, left panel), indicating that PDI and ERp57 exhibit different spectrums of substrates. Additionally, HRG was not detected in any of the ERp57-substrate complexes (Fig. 1b, central and right panels). Together, these results demonstrated that HRG was a substrate of PDI but not ERp57.

### PDI cleaves three disulfide bonds in HRG

Human HRG contains 16 Cys predicted to form 6 disulfide bonds[25]. To identify which disulfides were the targets of PDI, we employed differential Cys alkylation and mass spectrometry[26] (Fig. 1c) to quantify the redox state of Cys in HRG purified from human plasma. We detected 42 Cys-containing peptides that covered all 16 Cys in HRG (Supplementary Table S1) with high confidence of tandem mass spectra (Supplementary Fig. S2). Based on the redox state of these Cys, 8 rather than 6 disulfide bonds were predicted in human HRG. In the presence of 2- or 10-fold molar excess of PDI, 6 Cys were significantly more reduced: C306, C309, C390, C434, C409 and C410. These Cys form 3 disulfide bonds: C306–C309, C390–C434 and C409–C410 (Fig. 1d). The disulfide pairing of Cys were confirmed independently using a disulfide-linked peptide method[27]. Six of the 8 disulfide bonds were mapped (Supplementary Table S2) and their relative abundance to C6–C486 disulfide was calculated (Fig. 1e). C60–C71 and C87–C108 were cross-linked to form a tripeptide product that could not be resolved by MS/MS. Notably, all 3 disulfide bonds reduced by PDI reside in proximity to the histidine-rich region (HRR) (Fig. 1f). Cleavage of these disulfide bonds suggests a role for PDI in controlling interactions of HRR with its binding partners such as heparin, HS, cation $Zn^{2+}$ and FXIIa[28,29].

### PDI increases binding affinity of HRG with heparin and FXIIa

To characterize the consequence of PDI cleavage of HRG disulfide bonds[30], we first quantitatively measured the interactions of HRG with its ligands including heparin, an analog of endothelial HS, and FXIIa, using the ultra-sensitive biomembrane force probe (BFP)[31] (Fig. 2a). The molecular adhesion was detected based on the force trace between the Target bead (coated with heparin or FXIIa) and Probe bead (coated with HRG) during repetitive BFP cycles with each consisting of three consecutive steps: approach, contact and retraction, driven by a piezo actuator[32] (Supplementary Figs. S3a and S3b). The frequency of HRG adhesion was <2% when the target ligands (heparin or FXIIa) were absent on the beads (Supplementary Fig. S3c), confirming the binding specificity. Enumeration of the adhesion frequencies under different contact time showed that the interactions of HRG with heparin (Fig. 2b) and FXIIa (Fig. 2c) were both significantly increased in the presence of $Zn^{2+}$ (5 μM), consistent with previous reports[21,24], and these interactions were further enhanced when PDI-CCCC was added into the solution compared to PDI-AAAA. In addition, measurements of the binding kinetics showed that PDI-CCCC increased the cellular affinity[33] of HRG for heparin ($0.26 \pm 0.03 \text{ s}^{-1}$ $vs$ $1.87 \pm 0.19 \text{ s}^{-1}$, $P = 0.0024$) (Fig. 2d) and FXIIa ($0.36 \pm 0.06 \text{ s}^{-1}$ $vs$ $1.81 \pm 0.30 \text{ s}^{-1}$, $P = 0.0215$) (Fig. 2e) by 7- and 5-folds, respectively, while PDI-AAAA exhibited limited effects in the presence of $Zn^{2+}$. Taken together, these results showed that PDI cleavage of HRG disulfide bonds significantly enhanced the molecular interactions of HRG with heparin and FXIIa.

### PDI enhances HRG neutralization of HS by antithrombin displacement

We next determined the effect of PDI on the binding of purified human plasma HRG to heparin using the solid-phase binding assay. Compared to vehicle, treatment with PDI-CCCC, but not PDI-AAAA, markedly increased the binding of HRG to heparin-coated surface in a dose-dependent manner in the presence of $Zn^{2+}$ (1 μM) (Fig. 3a). However, this binding was completely abolished in the absence of $Zn^{2+}$, which is consistent with previous report[24]. In contrast, addition of $Ca^{2+}$ did not

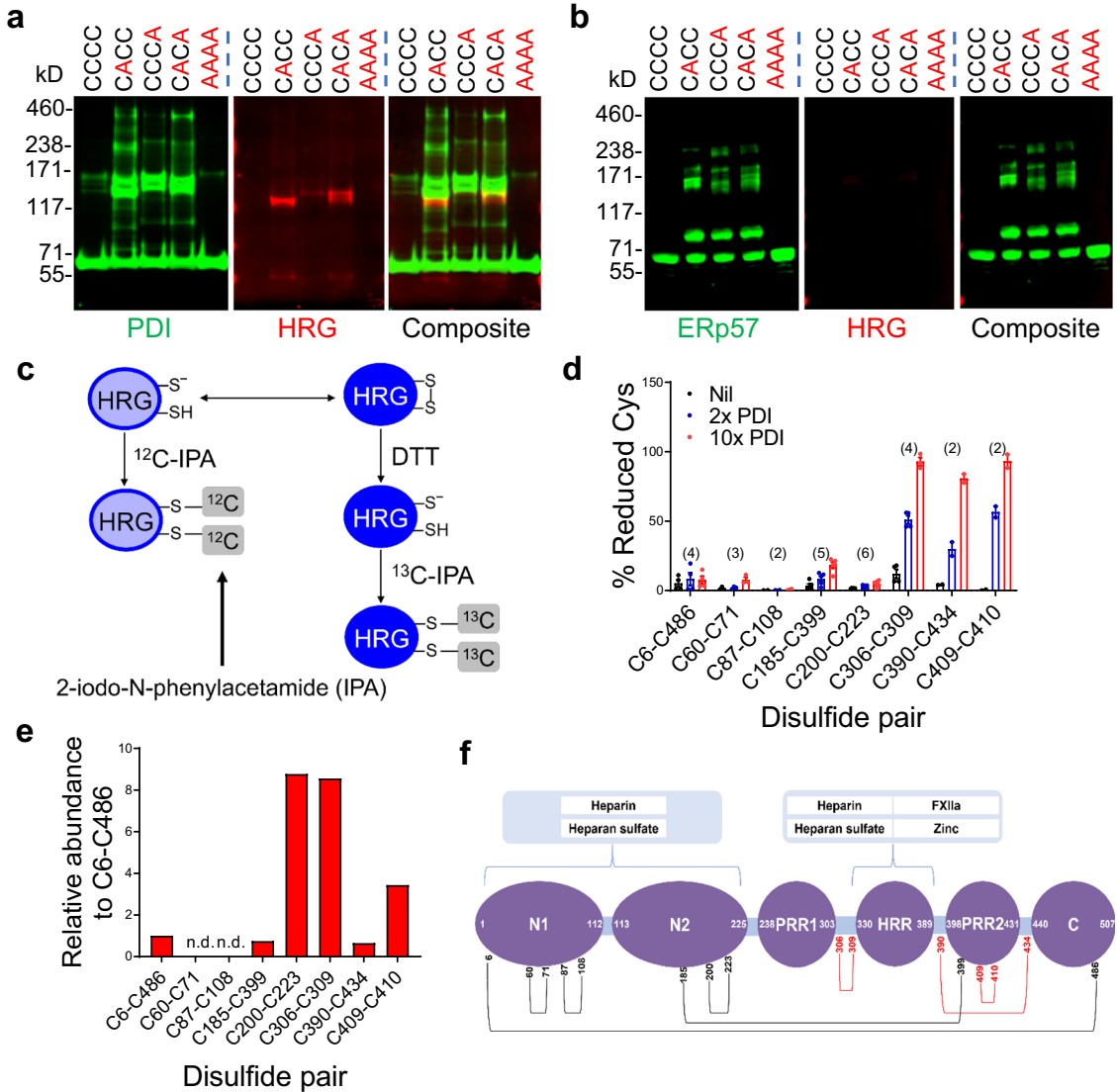

**Fig. 1 | PDI cleaves allosteric disulfide bonds in HRG.** Disulfide-linked complexes isolated from human platelet-rich plasma following mechanism-based kinetic trapping using variants of PDI (**a**) or ERp57 (**b**) were subjected to immunoblotting using anti-FLAG (Green) and anti-HRG antibody (Red) under non-reducing conditions. The representative blots from 3 independent experiments with similar results were shown. **c** The redox state of Cys was measured via differential alkylation using ¹²C-IPA and ¹³C-IPA (2-iodo-N-phenylacetamide) followed by chymotrypsin digestion and mass spectrometry analysis through a procedure as illustrated. **d** Redox state of Cys pairs was measured in purified human HRG incubated without (Nil) or with recombinant human PDI (2- or 10-fold molar excess). The number of peptides (*n* value) analyzed in each group was indicated above the bars. **e** The predicted

disulfide bonds were experimentally mapped using a method with disulfide-linked peptides and their relative abundance to the C6–C486 disulfide calculated. The cysteine pairing of the identified disulfide bonds in HRG was confirmed using label-free disulfide-linked peptides by mass spectrometry analysis. Except for C60–C71 and C87–C108 which crosslinked into a tripeptide with a molecular mass exceeding the limit of detection by the mass spectrometer, all disulfide bonds were mapped with relative abundance at a ratio of 0.3-8 as compared to the peptide containing C6–C486 bond. The data confirmed the identity of 3 disulfide bonds targeted by PDI. n.d., no peptide detected. **f** The disulfide pairs were located and mapped on the domain structure of human HRG with red color indicating the targets of PDI. Data are presented as mean values ± SEM. Source data are provided as a Source Data file.

increase the binding of HRG on heparin (Supplementary Fig. S4). HRG competes with antithrombin for the binding to vascular HS and neutralizes the anticoagulant activity of HS in a Zn²⁺-dependent manner during vascular injury[11,23,24]. We evaluated the effect of PDI on HRG binding to endothelial HS in cell culture. Human plasma pre-treated with PDI-CCCC had significantly increased binding of HRG concomitant with attenuated binding of antithrombin (Fig. 3b) on HUVECs as compared to PDI-AAAA-treated group. When HUVECs were pre-treated with heparanase, HRG binding was abolished (Supplementary Fig. S5), suggesting the interactions were glycosaminoglycans-dependent. To further quantitate the effect of PDI, we performed a cell-based ELISA assay. Similarly, addition of PDI-CCCC to plasma increased HRG binding (Fig. 3c) but decreased antithrombin binding

(Fig. 3d) to HUVECs in the presence of Zn²⁺. These combined results showed that PDI cleavage of allosteric disulfides in HRG led to enhanced HRG interactions with HS and displacement of antithrombin on endothelium.

### PDI enhances HRG inhibition of FXIIa protease activity

HRG also binds to FXIIa with high affinity through the HRR domain and inhibits the intrinsic pathway[21,22]. We determined the effect of PDI on the interaction of HRG with FXIIa. Purified human plasma HRG pre-treated with PDI-CCCC exhibited increased binding on FXIIa-coated surface in the presence of Zn²⁺ compared to PDI-AAAA- or vehicle-treated group (Fig. 3e). Additionally, increased HRG binding to FXIIa by PDI-CCCC led to enhanced inhibition of FXIIa activity also dependent

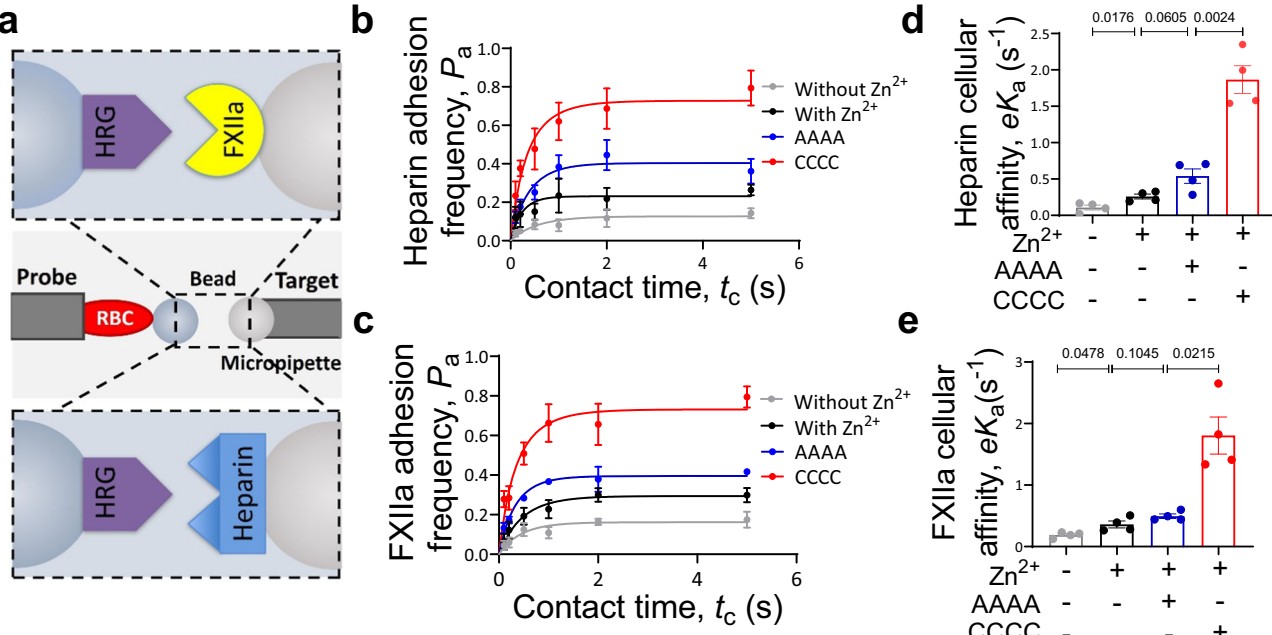

**Fig. 2 | PDI increases the binding of HRG with heparin and FXIIa.** HRG interactions with heparin and FXIIa were measured by a Biomembrane Force Probe (BFP). **a** The system used a micropipette to aspirate an engineered RBC with an HRG-coated Probe bead attached to its apex to form an ultrasensitive force sensor (middle). The Target bead coated with heparin (top) or FXIIa (bottom) was aspirated by an opposing micropipette to undergo a touch cycle consisting of approach, contact and retract steps with the Probe bead. The 'No adhesion' vs 'Adhesion' events were judged from the RBC's deflection and subsequently derived force spectroscopy traces. The adhesion frequency ($P_a$ = # 'Adhesion' / (# 'No adhesion' + # 'Adhesion')) between HRG and heparin (**b**) or FXIIa (**c**) was measured at different contact time ($t_c$) under indicated conditions: without $Zn^{2+}$, with $Zn^{2+}$, with inert-PDI (AAAA), or with wt-PDI (CCCC) in the presence of $Zn^{2+}$ (5 μM) ($n = 4$ probe-target pairs). The cellular affinity ($eK_a$) of HRG for heparin (**d**) and FXIIa (**e**) was derived by fitting the adhesion frequency ($P_a$) versus contact time ($t_c$) into the equation $P_a(t_c) = 1 - \exp\{-eK_a[1 - \exp(-k_{off}t_c)]\}$ ($n = 4$ probe-target pairs). The data are presented as mean values ± SEM and analyzed by Welch's ANOVA test (**d** and **e**). Source data are provided as a Source Data file.

on $Zn^{2+}$, as determined by cleavage of chromogenic substrate S2302 (Fig. 3f). To validate this observation in a physiological system, thrombin generation assay (TGA) was performed. Clotting reaction was initiated by the addition of FXIIa along with HRG pre-treated with vehicle, PDI-CCCC or PDI-AAAA into mouse plasma double deficient in FXII and HRG (*DKO*). There was no significant difference between vehicle- and PDI-AAAA-treated groups (Fig. 3g). Compared to PDI-AAAA, PDI-CCCC-treated HRG led to enhanced inhibition of FXIIa-induced TGA (Fig. 3g). Both the lag time and peak time were prolonged (Fig. 3h). The peak height and area under the curve (AUC) were also decreased (Fig. 3i). Alternatively, when TGA was initiated by activated partial thromboplastin time (aPTT) reagent using mouse plasma deficient in HRG (*Hrg⁻/⁻*) (Fig. 3j), supplement with recombinant mouse HRG significantly inhibited thrombin generation, which is consistent with previous report[21]. There was no difference between vehicle- and PDI-AAAA-treated HRG, whereas PDI-CCCC significantly augmented the inhibitory effect of HRG (Fig. 3k, l). To rule out the possibility that HRG interferes with other coagulation proteases, the activities of nonbinding enzymes such as thrombin and FXa were evaluated in the presence of different concentrations of purified HRG using respective substrates and no significant effect was detected (Supplementary Fig. S6). Taken together, these results showed that PDI, through its catalytic motifs, enhanced the inhibition of FXIIa activity and the intrinsic coagulation pathway by HRG.

**HRG incorporates into growing thrombi requiring platelet-derived $Zn^{2+}$**

We evaluated the in vivo incorporation of HRG into a growing thrombus by intravital microscopy. Using the laser injury-induced thrombosis model, we visualized the assembly of platelets and HRG using Dylight-649-conjugated anti-CD42c and Alexa-488-conjugated

anti-HRG, respectively (Fig. 4a; Supplementary Movie 1). No significant HRG signal was detected on the unperturbed cremaster arterioles prior to vessel injury (Fig. 4a). However, upon laser-induced injury and secretion of extracellular PDI[13,34], there was a rapid accumulation of fluorescence at the injury site in anti-HRG-infused mice compared to non-specific fluorescence in IgG-infused control mice (Fig. 4a). Analysis of the median fluorescent intensity over time demonstrated the rapid integration of HRG into a growing thrombus (Fig. 4b), which occurred prior to the major platelet peak (at ~80 s after vessel injury) (Fig. 4c). This finding indicated a role of HRG accumulation during the early phase of thrombosis.

Activated platelets release $Zn^{2+}$, a cofactor for HRG binding on HS, into the area of a growing thrombus[35,36]. We determined the contribution of platelets to HRG accumulation during thrombus formation. When mice were treated with eptifibatide, an integrin $\alpha_{IIb}\beta_3$ antagonist that eliminated platelet deposition (Fig. 4d) and thus platelet-derived $Zn^{2+}$, HRG accumulation at the site of vessel injury was significantly reduced by 58.3% ($P = 0.0001$) (Fig. 4e). These results suggested that HRG incorporated into growing thrombus upon vessel injury in a process requiring $Zn^{2+}$ secreted by activated platelets. Since HRG may bind to platelets in itself[37], to clearly determine the spatial distribution of HRG during thrombus formation, we performed in vivo two-photon imaging of the cremaster arterioles (Fig. 4f, marked as ROI1). Analysis of the focal planes showed that HRG (green) largely accumulated on vascular endothelium as labeled by CD31 (red) following laser-induced injury (Fig. 4f, marked as ROI2). The distribution of the respective intensity of green and red fluorescence along the indicated line that crossed the injury area exhibited a similar pattern that matched well spatially at the location of thrombus formation (Fig. 4g). The Pearson's correlation coefficients of the pixel intensity for HRG and CD31 was much higher in the thrombus (ROI2) than the

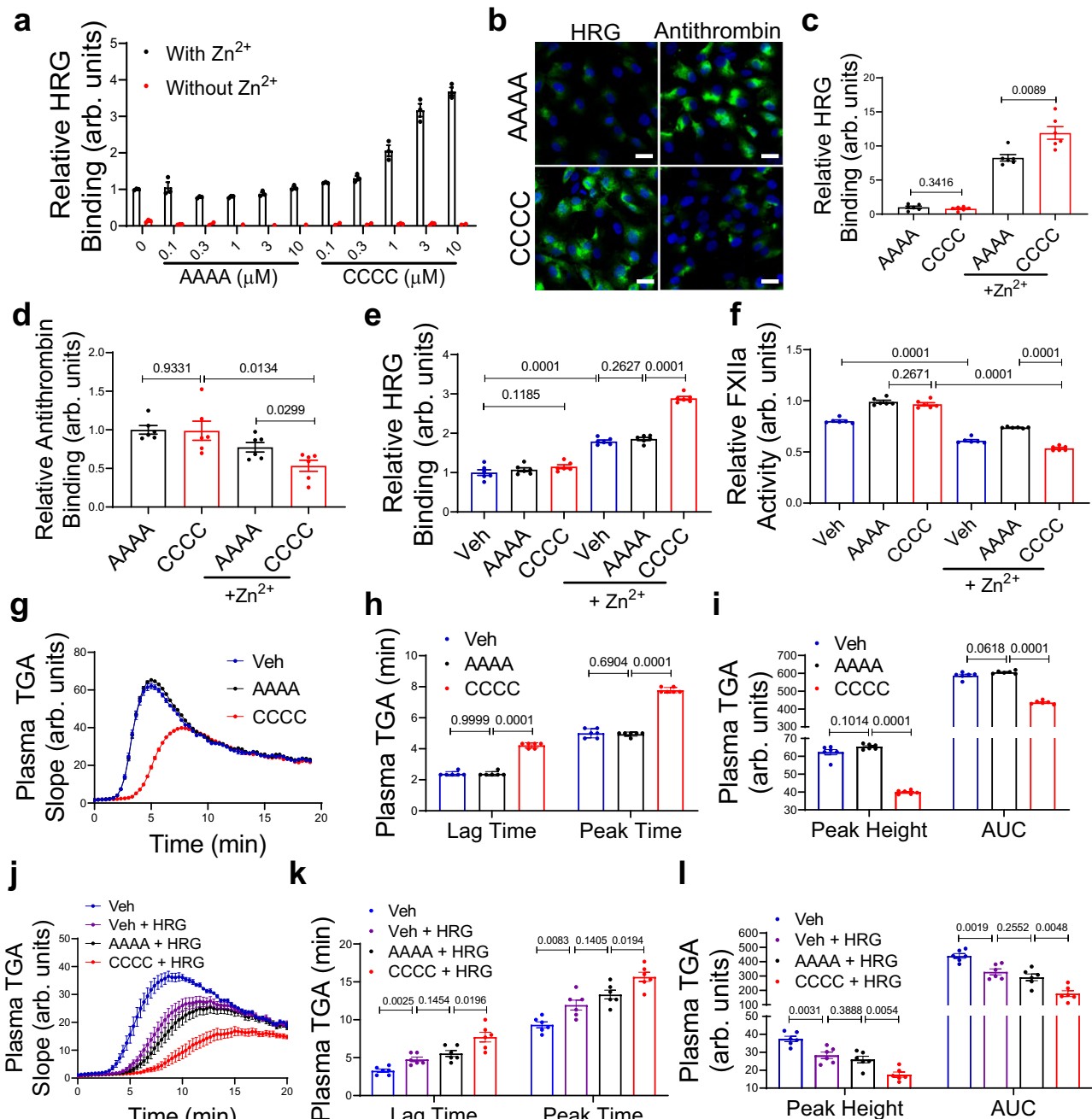

**Fig. 3 | PDI enhances the neutralization of heparan sulfate and inhibition of FXIIa by HRG. a** Purified human HRG was pre-treated with vehicle or recombinant human PDI variants at indicated concentrations and incubated on heparin-coated surface with or without $Zn^{2+}$ (1 μM). The binding of HRG was determined by ELISA ($n = 3$ independent samples). **b** Human plasma pre-treated with recombinant human PDI variants was incubated on HUVECs with $Zn^{2+}$ (10 μM). The binding of HRG and antithrombin on the cell surface were determined by immuno-fluorescence (scale bar: 20 μm). Human plasma pre-treated with recombinant human PDI variants was incubated on HUVECs with or without $Zn^{2+}$ (10 μM). The binding of HRG (**c**) and antithrombin (**d**) on the cell surface were determined by cell-based ELISA ($n = 6$ independent samples). **e** Purified human HRG was pre-treated with vehicle or recombinant human PDI variants and incubated on FXIIa-coated surface with or without $Zn^{2+}$ (1 μM). The binding of HRG was determined by ELISA ($n = 6$ independent samples). **f** Purified human HRG was pre-treated with

vehicle or recombinant human PDI variants, and mixed with FXIIa (10 nM) with or without $Zn^{2+}$ (1 μM). The activity of FXIIa was determined using chromogenic substrate S2302 ($n = 6$ independent samples). **g** Purified human HRG was pre-treated with vehicle or recombinant human PDI variants, mixed with FXIIa, and then added into *DKO* mouse plasma to initiate thrombin generation (TGA). The lag time and peak time (**h**), and the peak height and area under the curve (AUC) (**i**) from individual TGA curve were calculated ($n = 6$ independent samples). **j** Recombinant mouse HRG was pre-treated with vehicle or recombinant mouse PDI variants, and then added into *Hrg*$^{-/-}$ mouse plasma. TGA was initiated by addition of aPTT reagent. The lag time and peak time (**k**), and the peak height and AUC (**l**) from individual TGA curve were calculated ($n = 6$ independent samples). Veh vehicle. Data are presented as mean values ± SEM and analyzed by Welch's ANOVA test (**c**−**f**, **h**, **i**, **k** and **l**). Source data are provided as a Source Data file.

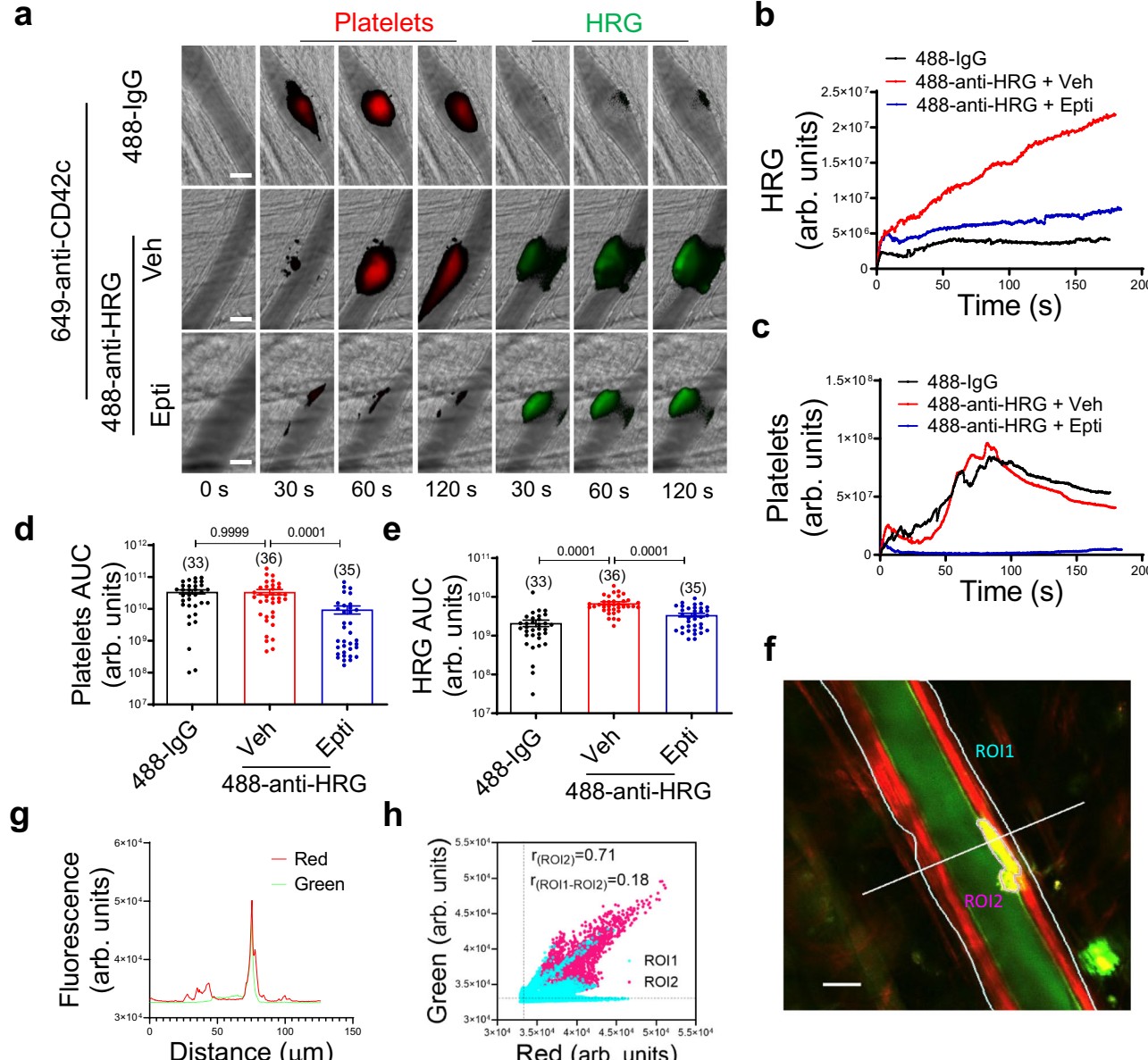

**Fig. 4 | HRG accumulates on the vessel wall during thrombus formation. a** The accumulation of platelets and HRG during laser injury-induced thrombus formation in murine cremaster arterioles were visualized using Dylight-649-congugated anti-CD42c (Red) and Alexa-488-conjugated anti-HRG (Green), respectively, with irrelevant IgG as background control (488-IgG) (scale bar: 30 μm) (Supplementary Movie 1). The median fluorescence intensity of HRG (**b**) and platelets (**c**) were calculated and plotted over time from mice treated or untreated with eptifibatide (10 μg/g body weight, repeated every 15-20 min). The area under the curve (AUC) for platelets (**d**) and HRG (**e**) were analyzed from each individual thrombus in different treatment groups as indicated. The number of thrombi (*n* value) analyzed in each group was indicated above the bars. **f** A representative focal plane of HRG accumulation and vascular endothelium, as detected by Alexa-488-conjugated anti-HRG (Green) and Alexa-647-conjugated anti-CD31 (Red), respectively,

approximately 5 min after laser-induced vessel injury, was extracted from in vivo two-photon scanning of the cremaster arterioles. The areas of the cremaster arteriole (ROI1, cyan) and the thrombus (ROI2, magenta) were manually defined (scale bar: 20 μm). **g** The intensities of green (HRG) and red (CD31) fluorescence were plotted spatially along the white solid line as indicated in **f**. **h** The scatter plot of pixel intensities of green (HRG) and red (CD31) fluorescence in ROI1 and ROI2 as indicated in **f** were presented. The Pearson's correlation coefficients (r) for pixels with intensity above the threshold (indicated by the dashed lines) in the thrombus (ROI2) and in the area without the thrombus (ROI1-ROI2) were inserted. Veh vehicle, Epti eptifibatide. Data are presented as mean values ± SEM and analyzed by Kruskal–Wallis test with Dunn's multiple comparisons (**d** and **e**). Source data are provided as a Source Data file.

rest of the arteriole (ROI1-ROI2) (Fig. 4h), indicating HRG largely localized at the interface between platelets and endothelium in the newly formed thrombus.

**Extracellular PDI regulates HRG accumulation during thrombus formation**

Based on the in vitro effect of PDI on HRG binding on endothelial cells, we determined the in vivo role of extracellular PDI in HRG

accumulation during thrombosis in mice. First, since human and murine HRG differ in their disulfide bond positions and patterns[38], we validated the effects of PDI on HRG using murine samples. Compared to PDI-AAAA, mouse plasma pre-treated with murine PDI-CCCC exhibited increased HRG binding on mouse endothelial cells (bEnd.3) as determined by immunofluorescence staining (Fig. 5a), confirming the interactions between PDI and HRG in mice. Second, in the FeCl₃-induced carotid thrombosis model when mice were treated

with quercetin-3-rutinoside (Rutin), an inhibitor that targets PDI but not other thiol isomerases including ERp57, ERp72 or ERp5[18], HRG accumulation at the site of vessel injury was blocked as determined by immunohistochemistry of thrombus cross sections (Fig. 5b). Third, in the laser injury-induced model of cremaster thrombosis (Fig. 5c; Supplementary Movie 2) inhibition of extracellular PDI by intravenous infusion of Rutin also suppressed the accumulation of HRG (Fig. 5d) by decreasing the fluorescent signal by 57.3% ($P = 0.0001$) based on analysis of the AUC of individual thrombus (Fig. 5e). This result was further confirmed using a synthetic small-molecule PDI inhibitor Bepristat 2a[39], which inhibited HRG accumulation to a similar extent as Rutin (Supplementary Fig. S7).

In a set of parallel experiments, we examined the incorporation of antithrombin during thrombus formation. In vitro treatment of mouse plasma with PDI-CCCC led to decreased binding of antithrombin on bEnd.3 cells (Fig. 5f). In vivo inhibition of extracellular PDI by Rutin in the FeCl₃-induced carotid thrombosis model resulted in increased antithrombin deposition at the injury site compared to vehicle-treated animals where marginal antithrombin binding on the vessel wall was observed during normal thrombosis (Fig. 5g). Further, in the laser-induced cremaster thrombosis model (Fig. 5h; Supplementary Movie 3) treatment with Rutin preserved the antithrombin signal (Fig. 5i) based on analysis of the AUC of individual thrombus (Fig. 5j). These combined results demonstrated that extracellular PDI enables HRG binding to endothelium and concurrently displacing antithrombin during thrombus formation.

### HRG has dual functions in modulation of the kinetics of thrombus formation

Our data indicated that HRG exerted a prothrombotic effect by displacement of antithrombin on HS. We determined the effect of HRG on antithrombin activity. Plasma from $Hrg^{-/-}$ mice exhibited significantly increased antithrombin activity on endothelial cell surfaces like in the presence of added heparin, which served as a positive control. In contrast, no significant antithrombin activity was detected in plasma per se (Fig. 6a). HRG also exerts antithrombotic effect through FXIIa. Notably, in the FeCl₃-induced thrombosis model, which is mediated in part by FXIIa[22], the antithrombotic activity of FXIIa-bound HRG[22] may counter-balance the prothrombotic activity of endothelial-bound HRG via its neutralization of HS[11,24,40,41]. To delineate the potential dual functions of HRG during thrombosis, we employed mice deficient in FXII ($F12^{-/-}$) and treated them with vivo-siRNA to knockdown plasma HRG (Fig. 6b). In the laser injury-induced thrombosis model (Fig. 6c; Supplementary Movie 4) initiated by TF[42], reduction of plasma HRG by siRNA attenuated platelet deposition (Fig. 6d) and fibrin generation (Fig. 6e) in $F12^{-/-}$ mice. Platelets and fibrin were reduced by 37.8% ($P = 0.0366$) (Fig. 6f) and 35.3% ($P = 0.0094$) (Fig. 6g), respectively.

These observations were further validated using HRG and FXII double-knockout (DKO) mice. Immunoblotting confirmed the absence of both FXII and HRG antigens in plasma from DKO mice (Fig. 7a). To determine the roles of HRG in coagulation in the presence and absence of FXII, the 4 groups of mice were evaluated using a modified TGA (Fig. 7b)[43] induced by TF on endothelial cells. Compared to WT, $Hrg^{-/-}$ plasma had enhanced but $F12^{-/-}$ plasma had decreased thrombin generation. Importantly, DKO plasma had further attenuated thrombin generation (Fig. 7b) with both reduced peak height (Fig. 7c) and AUC (Fig. 7d) compared to $F12^{-/-}$. These data suggest plasma HRG has a procoagulant role in the absence of FXII.

We next evaluated HRG functions during thrombosis using all 4 strains of mice in the FeCl₃-induced carotid artery thrombosis model. Consistent with previous reports[21,22], the time to arterial occlusion in $Hrg^{-/-}$ ($2.7 \pm 0.9$ min) was shorter than WT ($7.5 \pm 0.8$ min, $P = 0.0010$), confirming the antithrombotic role of HRG in this model. As expected, the occlusion time in $F12^{-/-}$ ($59.8 \pm 12.0$ min, $P = 0.0033$) was significantly prolonged. However, like the ex vivo experiments, DKO

exhibited further delay in arterial occlusion compared to $F12^{-/-}$ alone (Fig. 7e). These mice were additionally examined in the laser-induced injury model of thrombosis (Fig. 7f; Supplementary Movie 5). Compared to WT, platelet deposition (Fig. 7g) and fibrin generation (Fig. 7h) were enhanced in $Hrg^{-/-}$ mice but decreased in $F12^{-/-}$ mice consistent with previous studies[22,44]. However, DKO showed further inhibition of thrombus formation. Platelets and fibrin were decreased by 85.6% ($P = 0.0024$) (Fig. 7i) and 62.1% ($P = 0.0492$) (Fig. 7j) in DKO, respectively, compared to $F12^{-/-}$. Notably, analysis of the kinetics of thrombus formation showed that HRG deficiency led to delayed onset of fibrin generation (Fig. 7h) in $Hrg^{-/-}$ and DKO mice compared to WT and $F12^{-/-}$ animals as shown in the early phase of thrombus formation (e.g. 15 sec post vessel injury) in the representative images (Fig. 7f). These combined results demonstrate both the prothrombotic and antithrombotic functions of HRG during thrombosis, with the antithrombotic influence taking precedence.

### A gain-of-function (gof) mutant of HRG inhibits thrombosis

Murine HRG has equal number of total amino acids and Cys residues as in human with conserved domain arrangement across different species[25,38]. Particularly, the Cys residues for the disulfide bonds within the N-terminal domains (C60–C71, C85–C106 and C198–C221), between the N-terminal and the C-terminal domains (C6–C486), and between the N-terminal and PRR2 domains (C183–C396), are conserved in mouse HRG[38]. Based on the location of disulfide bonds that are targeted by PDI in human HRG (Fig. 1f), we predicted that PDI cleaves 3 disulfide bonds on murine HRG: C246–C251, C270–C467 and C315–C320 corresponding to C306–C309, C390–C434 and C409–C410 in human. These bonds were also located in proximity to the HRR domain (Fig. 8a). Mutation of the 6 Cys to Ala prevented the formation of the potential disulfide bonds and locked murine HRG in a form equivalent to PDI reduction, leading to gain of functions (gof-HRG) that are regulated by PDI. These murine HRG variants were expressed in HEK293 cells and evaluated both in vitro and in vivo. Compared to wt-HRG, gof-HRG exhibited increased binding both on heparin (Fig. 8b) and FXIIa (Fig. 8c). Accordingly, when labeled with fluorophore and infused into $Hrg^{-/-}$ mice, gof-HRG showed increased accumulation (Fig. 8d, e) at the site of vessel injury in the laser-induced thrombosis model. Further, compared to wt-HRG, reconstitution with gof-HRG in $Hrg^{-/-}$ mice led to reduced platelet accumulation (Fig. 8f) and fibrin generation (Fig. 8g) based on the analysis of AUC of individual thrombus (Fig. 8h, i). These data confirmed that the target disulfide bonds of PDI allosterically regulates the functions of HRG during coagulation.

## Discussion

This investigation shows that extracellular PDI cleaves 3 allosteric disulfide bonds C306–C309, C390–C434 and C409–C410 in human HRG. Reduction of these disulfides enhances the interactions of HRG with its binding partners including HS and FXIIa. In the presence of $Zn^{2+}$ presumably released by the bound platelets following vessel injury, PDI reduces HRG to neutralize endothelial HS via displacement of antithrombin allowing for accelerated activation of the clotting system. On the other hand, PDI also reduces HRG to enhance its inhibition of FXIIa to prevent excessive clot formation (Fig. 9). These previously unappreciated regulatory pathways working simultaneously, but in apparent opposite directions represent a novel mechanism whereby extracellular PDI fine-tunes thrombus formation by modifying allosteric disulfides of plasma HRG. These findings advance our understanding of the biology of clotting disorders.

Allosteric disulfide bonds targeted by extracellular PDI have been identified in several proteins that are involved in thrombosis[15,17,30], including platelet surface receptors[45,46], coagulation factors[47] and extracellular matrix proteins[20]. We previously showed that cleavage of two disulfide bonds in plasma vitronectin, C137-C161 and C274-C453,

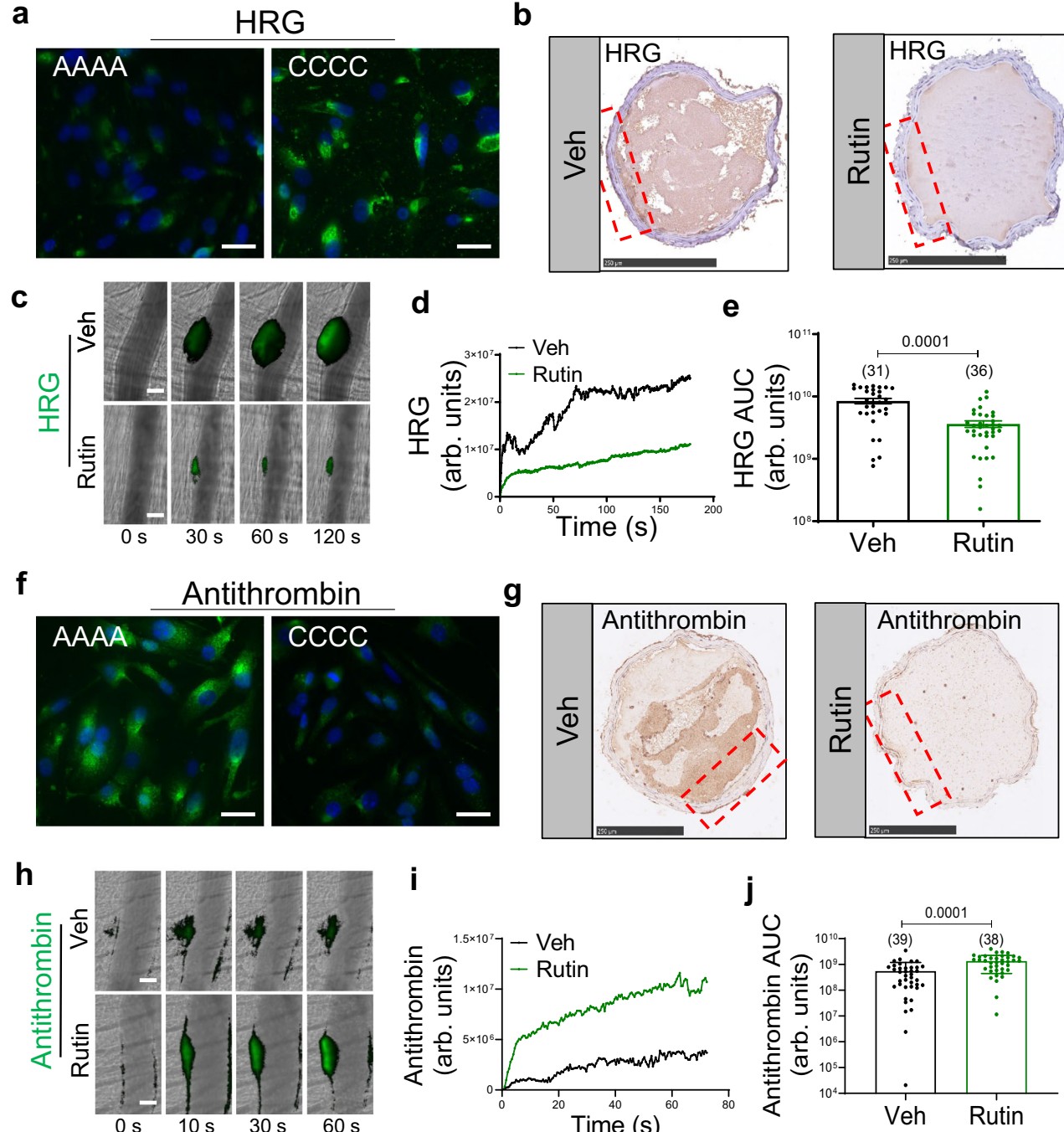

**Fig. 5 | PDI regulates the incorporation of HRG and antithrombin during thrombus formation. a** Mouse plasma pre-treated with recombinant mouse PDI-CCCC or PDI-AAAA was incubated on mouse cerebral microvascular endothelial cells (bEnd.3) in the presence of $Zn^{2+}$ (100 μM). The binding of HRG on the cell surface were determined by immunofluorescence using Alexa-488-conjugated antibodies (scale bar: 20 μm). **b** Representative images of immunohistochemistry of HRG from cross sections of thrombus induced by $FeCl_3$ in the carotid artery in mice treated with vehicle or Rutin (scale bar: 250 μm). **c** Representative images of HRG incorporation (Green), visualized using Alexa-488-conjugated anti-HRG, at indicated time points following laser injury in the cremaster arterioles in mice treated with vehicle or Rutin (5 μg/g body weight) (scale bar: 30 μm) (Supplementary Movie 2). The median fluorescence intensity (**d**) and the area under the curve (AUC) (**e**) of HRG were analyzed from each individual thrombus in the two groups. The number of thrombi (n value) analyzed in each group was indicated above the bars. **f** Mouse plasma pre-treated with recombinant mouse PDI-CCCC or

PDI-AAAA was incubated on bEnd.3 cells in the presence of $Zn^{2+}$ (100 μM). The binding of antithrombin on the cell surface were determined by immunofluorescence using Alexa-488-conjugated antibodies (scale bar: 20 μm). **g** Representative images of immunohistochemistry of antithrombin from cross sections of thrombus induced by $FeCl_3$ in the carotid artery in mice treated with vehicle or Rutin (scale bar: 250 μm). **h** Representative images of antithrombin incorporation (Green), visualized using Alexa-488-conjugated anti-antithrombin, at indicated time points following laser injury in the cremaster arterioles in mice treated with vehicle or Rutin (5 μg/g body weight) (scale bar: 30 μm) (Supplementary Movie 3). The median fluorescence intensity (**i**) and AUC (**j**) of antithrombin were analyzed from each individual thrombus in the two groups. The number of thrombi (n value) analyzed in each group was indicated above the bars. Veh vehicle, Rutin quercetin-3-rutinoside. Data are presented as mean values ± SEM and analyzed by two-tailed Mann–Whitney $U$-test (**e** and **j**). Source data are provided as a Source Data file.

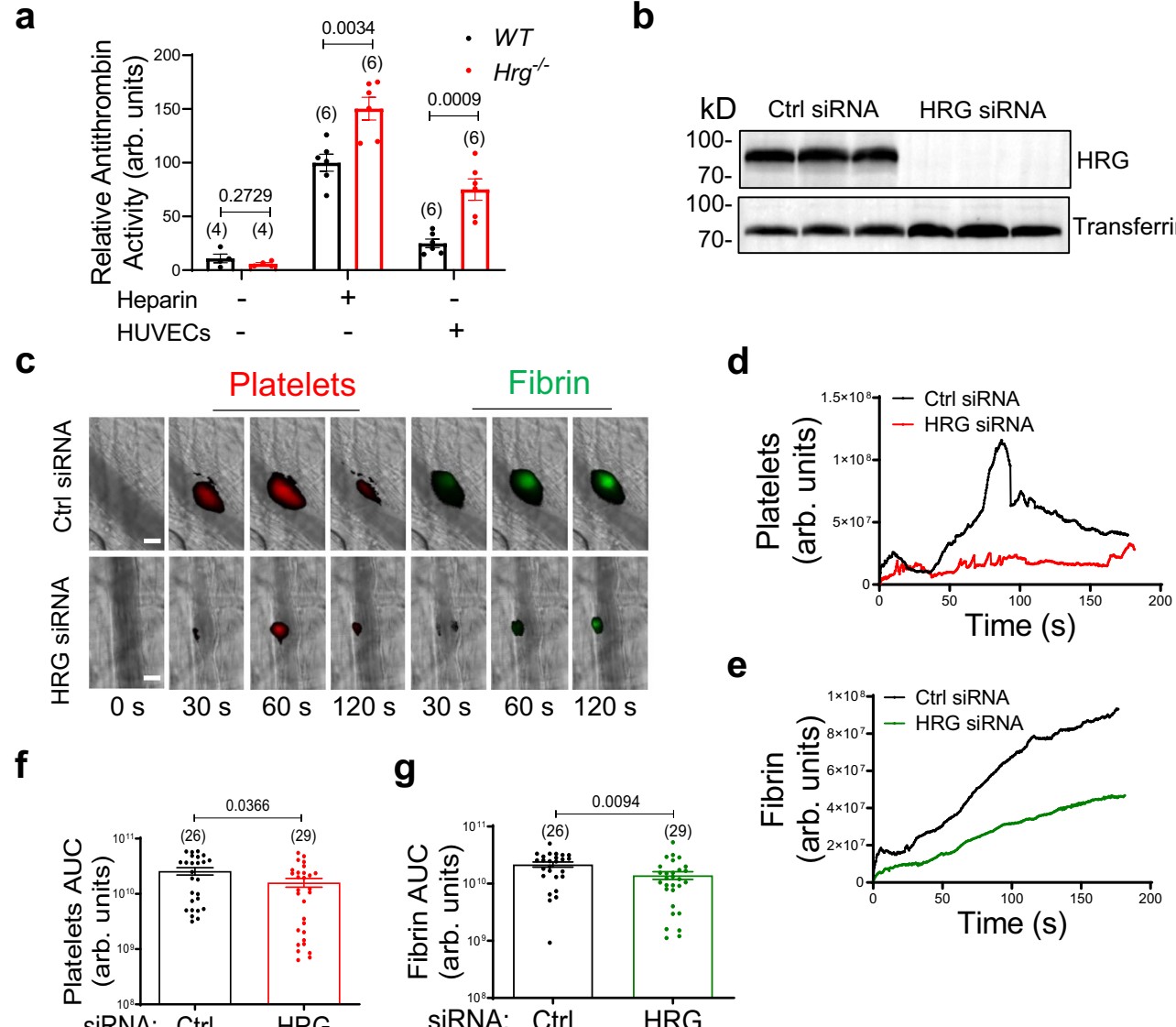

**Fig. 6 | Plasma HRG exhibits prothrombotic roles in the absence of FXII. a** The activity of antithrombin was measured in *WT* and *Hrg⁻/⁻* plasma on endothelial cell surface, or in the presence or absence of heparin. The number of independent samples (n value) analyzed in each group was indicated above the bars. **b** Immunoblotting of HRG antigen in equal amounts of plasma from *F12⁻/⁻* mice treated with control or vivo-siRNA to HRG, with transferrin as the loading control. **c** Representative images of platelet accumulation (Red) and fibrin generation (Green), as visualized by Dylight-649-conjugated anti-CD42c and Alexa-488-conjugated 59D8 antibody, respectively, at indicated time points following laser injury in the cremaster arterioles in *F12⁻/⁻* mice treated with control or vivo-siRNA to HRG (scale bar: 30 µm) (Supplementary Movie 4). The median fluorescence intensity of platelets (**d**) and fibrin (**e**) were calculated and plotted over time. The area under the curve (AUC) for platelets (**f**) and fibrin (**g**) were analyzed from each individual thrombus in the two groups. The number of thrombi (*n* value) analyzed in each group was indicated above the bars. Ctrl, control. Data are presented as mean values ± SEM and analyzed by two-tailed Welch's *t*-test (**a**) or two-tailed Mann–Whitney *U*-test (**f** and **g**). Source data are provided as a Source Data file.

by vascular PDI enables the binding of vitronectin to β3 integrins during thrombus formation[20]. Li et al. demonstrated that surface-bound PDI regulates the ligand-binding activity of glycoprotein Ibα (GPIbα) by reducing two allosteric disulfide bonds C4-C17 and C209-C248 in the receptor[45]. In the current study, we determined that three target disulfides in plasma HRG are proximal to the HRR domain. It is speculated that these disulfide bonds lock native plasma HRG in a cryptic form preventing certain interactions under physiologic circumstances. However, upon vessel injury, PDI is secreted to deactivate the clamp by cleaving these allosteric bonds, which exposes the functional HRR domain and enables HRG interactions with its binding partners including HS and FXIIa. This event influences the kinetics of thrombus formation. These combined studies suggest that regulation of allosteric disulfide bonds by extracellular PDI represents a general

mechanism whereby the intravascular release of PDI and other thiol isomerases serve as a "master switch" to prime thrombus formation.

We showed that PDI functions as a reductase on plasma HRG during thrombosis. Currently there are two proposed models that would allow PDI reductase activity in the extracellular space[30]. First, PDI may act through a redox chain involving members of the vascular thiol isomerase family which work cooperatively by shuffling elections among themselves and to substrate disulfides[13]. This model is based on the evidence that these thiol isomerases can exchange electrons with each other[48]. Second, PDI may function through a single-turnover manner, i.e., consumed as a reductant secreted into the extracellular milieu to cleave one substrate without further reduction[30]. This model is supported by extracellular burst of PDI by activated platelets and endothelial cells during thrombosis[34,49]. Although in both scenarios the

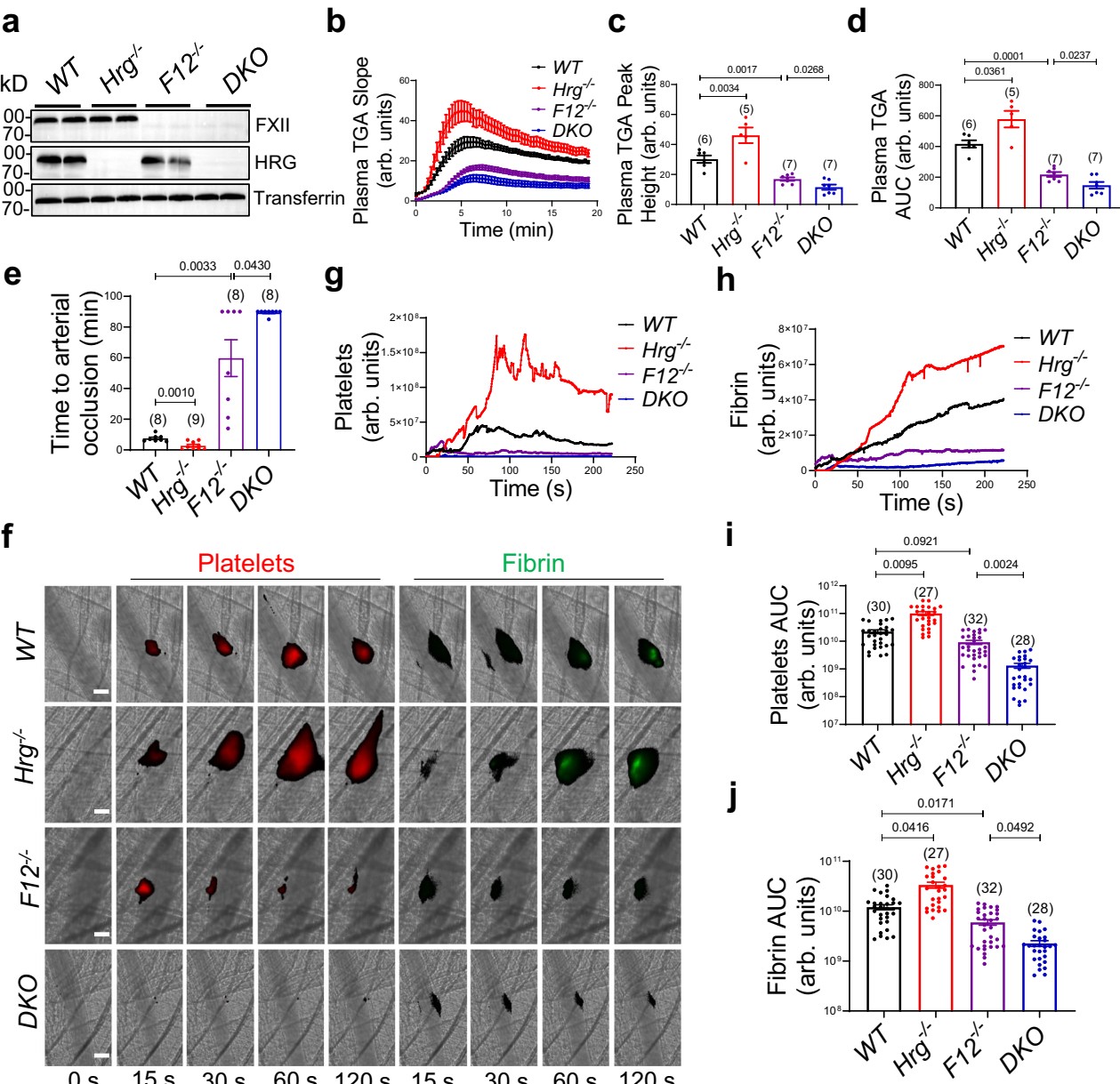

**Fig. 7 | Plasma HRG differentially regulates thrombus formation.**
**a** Immunoblotting of HRG and FXII antigens in equal amounts of plasma from 4 groups of mice as indicated: *WT*, *Hrg⁻/⁻*, *F12⁻/⁻* and *DKO*, with transferrin as the loading control. **b** Tissue factor-induced thrombin generation (TGA) in the plasma from 4 groups of mice was evaluated in the presence of endothelial cells. The peak height (**c**) and area under the curve (AUC) (**d**) from individual TGA curve were calculated in these groups as indicated. The number of animals (n value) analyzed in each group was indicated above the bars. **e** The time to complete arterial occlusion following FeCl₃-induced carotid injury were recorded in 4 groups of mice as indicated. The number of animals (n value) analyzed in each group was indicated above the bars. **f** Representative images of platelet accumulation (Red) and fibrin

generation (Green), visualized by Dylight-649-conjugated anti-CD42c and Alexa-488-conjugated 59D8 antibody, respectively, at indicated time points following laser injury in the cremaster arterioles in 4 groups of mice as indicated (scale bar: 30 μm) (Supplementary Movie 5). The median fluorescence intensity of platelets (**g**) and fibrin (**h**) were calculated and plotted over time. The AUC for platelets (**i**) and fibrin (**j**) were analyzed from each individual thrombus in different groups. The number of thrombi (n value) analyzed in each group was indicated above the bars. Data are presented as mean values ± SEM and analyzed by Welch's ANOVA test (**c–e**) or Kruskal–Wallis test with Dunn's multiple comparisons (**i** and **j**). Source data are provided as a Source Data file.

original sources that regenerate electrons to fuel the reactions on the cell surface are unknown, it is more likely that secreted PDI is pre-reduced by thioredoxin and/or NADPH in a segregated intracellular compartment[17].

Previous research predicted 6 disulfide bonds in human HRG based on the study in bovine protein[25,50]. However, our results showed there were 8 rather than 6 disulfide bonds in human HRG, which was independently confirmed using a disulfide-linked peptide method (Fig. 1)[27]. Since the domain structure and the pattern of disulfide bond

arrangement of HRG are conserved among species[25], the discrepancy in disulfide bond number may be attributed to the relatively low sequence identity (63%) of HRG protein between the 2 species, with 2 more Cys residues in human protein than bovine. Notably, murine HRG has the same number of total amino acids and Cys residues as in human, and is predicted to contain a total of 8 disulfide bonds as well. Analysis of the sequence homology between mouse and human showed that 5 out of the 8 disulfide bonds (i.e. those within the N-terminal domains, between the N-terminal and the C-terminal

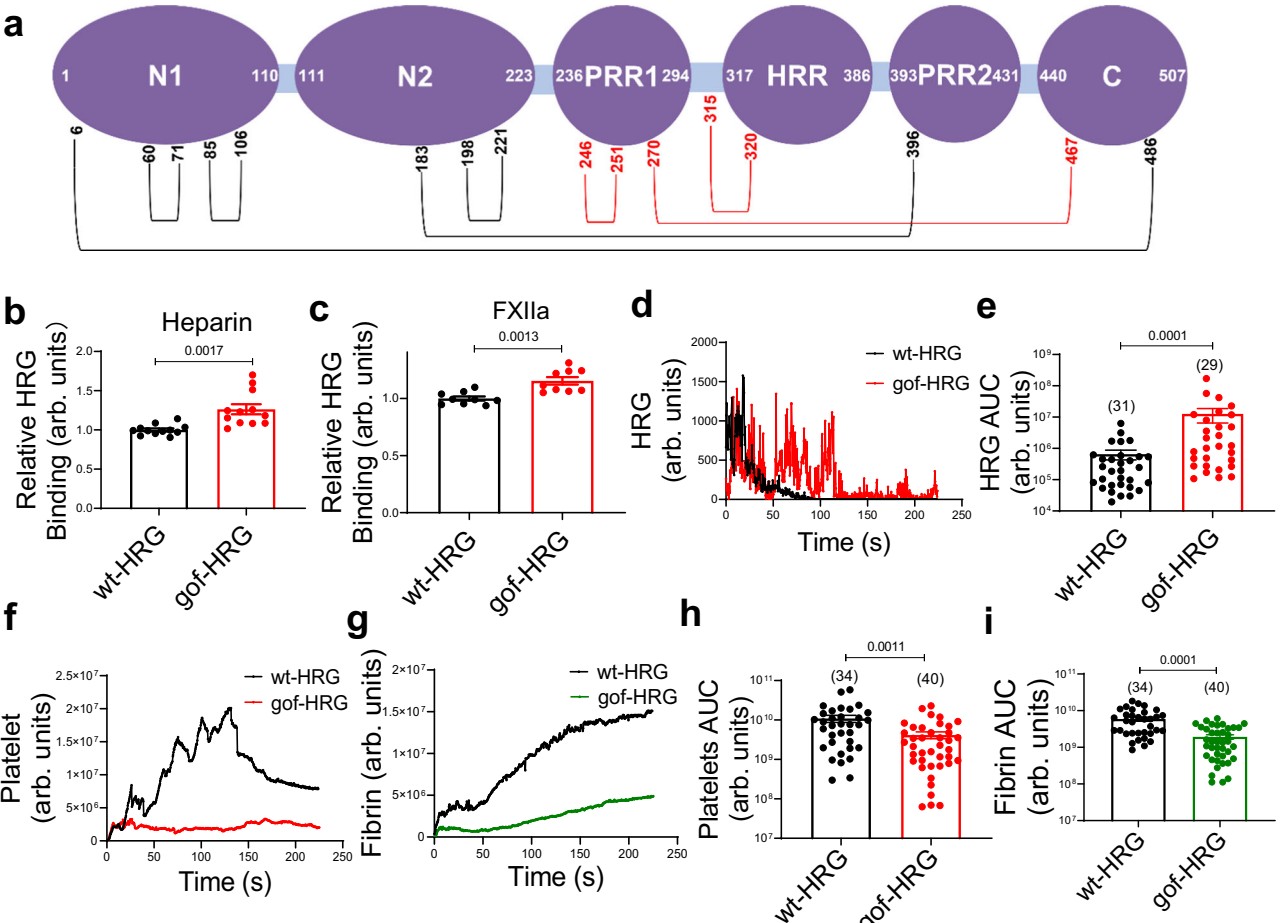

**Fig. 8 | A gain-of-function mutant of HRG reduces thrombus formation. a** The disulfide pairs were predicted on the domain structure of mouse HRG based on that of human HRG, with red color indicating the targets of PDI. The gain-of-function mutant of mouse HRG (gof-HRG) was generated by replacing 6 Cys of the 3 target disulfide bonds of PDI: C246-C261, C270–C467 and C315–C320 to Ala, which locked the variant in a reduced form. The binding of wt-HRG and gof-HRG on heparin (**b**) ($n = 12$ independent samples) and FXIIa (**c**) ($n = 9$ independent samples) was determined by ELISA. **d, e** The accumulation of HRG following laser injury in the cremaster arterioles of $Hrg^{-/-}$ mice infused with different HRG variants were visualized using Alexa-488-conjugated anti-HRG. The median fluorescence intensity of HRG (**d**) was plotted over time. The area under the curve (AUC) for HRG (**e**) was analyzed from each individual thrombus in different groups. The number of thrombi (n value) analyzed in each group was indicated above the bars. **f–i** Platelet accumulation and fibrin generation following laser injury in the cremaster arterioles of $Hrg^{-/-}$ mice infused with different HRG variants were visualized by Dylight-649-congugated anti-CD42c and Alexa-488-conjugated 59D8 antibody, respectively. The median fluorescence intensity of platelets (**f**) and fibrin (**g**) were calculated and plotted over time. The AUC for platelets (**h**) and fibrin (**i**) were analyzed from each individual thrombus in different groups. The number of thrombi (n value) analyzed in each group was indicated above the bars. Data are presented as mean values ± SEM and analyzed by two-tailed Welch's $t$-test (**b** and **c**) or two-tailed Mann–Whitney $U$-test (**e**, **h** and **i**). Source data are provided as a Source Data file.

domains, and between the N-terminal and PRR2 domains), are composed of conserved Cys residues[38], possibly in the form of structural disulfide bonds. Interestingly, the remaining 3 disulfide bonds, both in human and murine protein, are the targets of PDI reduction and allosterically regulate HRG functions. Such coincidence reflects the nature of difference between allosteric and structural disulfide bonds, with the latter one usually inert and evolutionarily conserved for the stability of protein architecture[30]. In the ex vivo coagulation assay (Fig. 2) to investigate the role of PDI in the inhibitory effect of HRG on FXIIa activity, thrombin generation was initiated by the addition of a mixture of human proteins including PDI, HRG and FXIIa into *DKO* mouse plasma. Even though the proteins were from different species, the fact that PDI, HRG and FXIIa were all from human allows for the proper biochemical reactions between them. Therefore, the sequence difference between human HRG and murine HRG does not compromise the conclusion of this experiment.

There was significant HRG accumulation during thrombus formation, which was partially reduced by eptifibatide. However, it is less

likely that the effect of eptifibatide is due to reduction of platelet-derived HRG[36] or the retention of fibrin-derived HRG[51], if any in vivo. First, the data from scanning two-photon intravital microscope showed that PDI-HRG functions at the interface between platelet thrombi and vessel wall. This location is consistent with the architectural characteristics of a growing thrombus in the laser-induced injury model[52]. In this model granule exocytosis, thrombin generation and fibrin deposition are restricted to the core region where HRG interacts with HS and FXIIa in a process dependent on PDI and $Zn^{2+}$ released by activated platelets and endothelial cells. Thus, the reduction of HRG accumulation in the presence of eptifibatide is not a consequence of elimination of HRG derived from platelet thrombi, but more likely due to reduced availability of platelet-derived $Zn^{2+}$ in the local milieu. Second, in the laser-induced injury model of thrombosis treatment with eptifibatide eliminates platelet accumulation without significantly affecting fibrin generation[49,53,54]. Hence, the unaltered fibrin generation but decreased HRG accumulation by eptifibatide excludes a major role of fibrin in HRG accumulation during

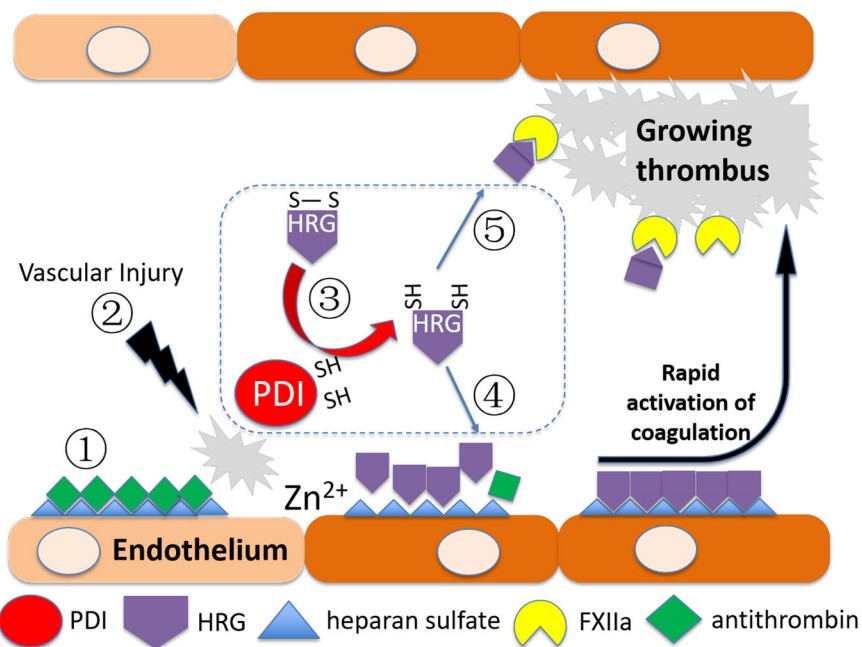

**Fig. 9 | A mechanism mediated by allosteric disulfide bonds fine-tunes thrombus formation. 1.** Under physiological condition, antithrombin binds to vascular heparan sulfate (HS) to inhibit the activation of coagulation; **2.** Vessel injury induces the release of $Zn^{2+}$ and PDI from activated platelets and endothelial cells; **3.** Extracellular PDI cleaves allosteric disulfide bonds in plasma HRG; **4.** Reduced HRG by PDI binds to endothelial HS to displace antithrombin, promoting rapid activation of coagulation. **5.** Reduced HRG by PDI also binds to FXIIa during thrombus growth, preventing excessive clot formation.

thrombosis. In fact, eptifibatide blocks integrin αIIbβ3 and thus platelet-platelet interactions without affecting the initial single layer of integrin-independent platelet adhesion at the core region following vessel injury[55]. The current study showed that $Zn^{2+}$ and PDI function in a synergistical manner with $Zn^{2+}$ alone sufficient to support HRG binding to endothelium, albeit at basal level, whereas PDI markedly potentiated the binding. Therefore, when treated with eptifibatide, the ambient $Zn^{2+}$ secreted by the remaining layer of platelet adhesion and possibly PDI released by activated endothelial cells at the local milieu allow HRG binding at reduced level, as evidenced by the partial reduction of HRG accumulation by eptifibatide. However, when treated with Rutin or Bepristat 2a, the PDI inhibitors[18,39] to block platelet accumulation (and therefore the burst of secreted $Zn^{2+}$) and extracellular PDI secreted by injured endothelial cells, further reduction of HRG accumulation was observed.

HRG is involved in thrombosis possibly by two mechanisms: (1) HRG neutralization of the anticoagulant HS may represent an important mechanism by which blood clotting is rapidly initiated[11,23]; (2) Inhibition of FXIIa by HRG suppresses intrinsic coagulation cascade, therefore putting the "brakes" on clot formation[8]. In accordance with these dual functions, $Hrg^{-/-}$ mice have (1) increased plasma antithrombin activity[56] and (2) accelerated $FeCl_3$-induced carotid artery thrombus formation[22]. Current understanding about the pathophysiological significance of HRG accumulation during thrombus formation are based on its interactions with FXIIa. In this study we revealed the dual roles of HRG during thrombosis. The fact that the absence of HRG enhanced thrombosis suggests that the role of FXIIa and the contact pathway is important for thrombosis and supersedes the effect of HRG on HS neutralization. Thus, PDI inhibition may result in diminution of the overall antithrombotic effect of HRG. However, it should be noted that PDI, as a "master switch"[13], also acts on other substrates in addition to HRG. Inhibition of PDI led to attenuated vitronectin accumulation, reduced platelet aggregation and diminished fibrin generation[17,20]. Therefore, the overall effect of PDI inhibitors, depending on the spectrum of substrates, is antithrombotic.

Nevertheless, given the anticoagulant effect of HRG, suppression of HRG by PDI inhibitors provides a procoagulant potential which puts a "brake" on the potent antithrombotic effects caused by PDI inhibition. Thus, the PDI-HRG pathway may provide mechanistic insights for the clinical effects of PDI inhibitors such as Rutin and Isoquercetin, which inhibit thrombosis without causing significant bleeding[18,19].

The utilization of intravital microscopy allowed for the examination of the kinetics of in vivo thrombus formation. During the initiation phase of thrombosis, triggered by TF of the extrinsic pathway, HRG neutralizes vascular HS and possibly other anticoagulant glycosaminoglycans to promote the accelerated activation of the coagulation system. This is evidenced by delayed onset of fibrin generation in $Hrg^{-/-}$ and $DKO$ as compared to $WT$ and $F12^{-/-}$. During the propagation of thrombosis, mediated by FXIIa of the intrinsic pathway, HRG inhibits FXIIa to suppress excessive thrombus growth, as evidenced by enhanced total platelet deposition and fibrin generation in $Hrg^{-/-}$ compared to $WT$ mice. Therefore, the utilization of $F12^{-/-}$ unmasked the potential procoagulant roles of HRG independent of the intrinsic coagulation pathway. Such characteristics of HRG are consistent with its multi-faceted functions as an adaptor protein in the plasma[29], and highlight its critical roles during the fine-tuning of thrombus formation. It should be noted that there are multiple heparin binding proteins in the plasma. Therefore, the neutralization of vascular HS by HRG may not be the sole determinant but a contributing factor for the rapid initiation of coagulation. For instance, vitronectin, a PDI substrate we previously identified[20], also binds to glycosaminoglycan to neutralize vascular anticoagulant functions[11,57]. Accordingly, vitronectin-deficient mice exhibited reduced fibrin generation during thrombus formation[20]. Given the critical roles of PDI in thrombosis, PDI inhibitors are being evaluated in thrombotic disorders. For instance, isoquercetin[18] has been shown to improve coagulation markers in advanced cancer patients in a phase-II clinical trial for the treatment of cancer-associated thrombosis[19]. Our results will contribute to precisely understanding the underlying mechanisms of these PDI inhibitors as novel antithrombotic agents. Further, the discovery of a gain-of-

function mutant of HRG (gof-HRG) by mutating the disulfide bonds targeted by PDI not only confirmed the specificity of the reaction between the two proteins, but also provides a strategy for the design of antithrombotic agent by targeting the intrinsic coagulation pathway.

In conclusion, we showed that extracellular PDI cleaves allosteric disulfide bonds in HRG, which enhances its binding to HS and FXIIa. The PDI-HRG pathway influences the development of thrombosis by two mechanisms: (1) promoting the rapid initiation of thrombosis mediated by TF via displacing antithrombin on the endothelium; (2) preventing excessive clot formation via inhibition of FXIIa-mediated propagation of coagulation. The PDI-HRG interactions present an underappreciated mechanism whereby extracellular PDI fine-tunes thrombus formation via modification of allosteric disulfides. This study provides mechanistic insights for the clinical effects of PDI inhibitors which reduce thrombosis without promoting bleeding[18,19].

## Methods

### Ethical regulations
This study complies with all relevant ethical regulations. The animal procedures were approved by the Institutional Animal Care and Use Committee of Huazhong University of Science and Technology (HUST), and in accordance with the Guide for the Care and Use of Laboratory Animals promulgated by the National Institutes of Health. The procedures involving human samples including isolation of pooled normal plasma and preparation of primary human umbilical vein endothelial cells (HUVECs) were approved by the Ethics Review Board of Tongji Medical College, HUST, with informed consent obtained from the healthy volunteers. This study did not collect or report the information on sex and gender of the volunteers. No sex-based analysis was included in the study design.

### Animals
FXII-deficient mice ($F12^{-/-}$) were described in a previous study[58]. HRG-deficient mice ($Hrg^{-/-}$) were generously provided by Dr. Wilhelm Jahnen-Dechent (RWTH Aachen University, Germany). FXII and HRG double-knockout mice (DKO) were generated by cross breeding $F12^{-/-}$ with $Hrg^{-/-}$. All mice were on a C57BL/6 background. C57BL/6 mice were purchased from Charles River (Beijing, China). The animals were housed in a standard facility with ambient temperature at 20–23 °C and humidity at 30–60% with 12 h light/dark cycles.

### Purification of human and mouse PDI variants
The cDNA of human PDI variants: CCCC (WT), CACC (C56A), CCCA (C400A), CACA (C56A, C400A) and AAAA (C53A, C56A, C397A, C400A), and human ERp57 variants, both in the pT7-Flag-SBP vector (Sigma), were generously provided by Dr. Bruce Furie of Harvard Medical School. The vectors were expressed in *E.coli* and purified using streptavidin-agarose beads (Pierce) as previously described[20]. The mouse PDI variants: CCCC (WT) and AAAA (C55A, C58A, C399A, C402A) were synthesized by SinoBiological (Beijing, China), cloned into pET15b vector, and purified using Ni-NTA His Bind Resin Beads (7Sea Biotech, Shanghai, China). The enzymatic activity of these variants were confirmed using insulin reduction assay[59].

The cDNA of mouse HRG variants: WT and gof-HRG (C246A, C251A, C270A, C315A, C320A, C467A) were synthesized by SinoBiological, cloned into the pSTEP2 vector with a C-terminal His tag. and transiently expressed in HEK293F cells. The recombinant proteins in the supernatant were collected and purified using Ni-NTA His Bind Resin Beads (7Sea Biotech, Shanghai, China).

### Cell culture
Primary human umbilical vein endothelial cells (HUVECs) were prepared from umbilical cords and characterized as previously described[54]. Primary HUVECs were cultured using endothelial cell medium (ECM, ScienCell) with heparin added (0.1 mg/mL). Passages

1–6 were used for experiments. Mouse cerebral microvascular endothelial cells (bEnd.3) were purchased from AnweiSci (Shanghai, China) and cultured using DMEM (10% FBS) with high glucose and low bicarbonate (1.5 g/L).

### Purification of HRG from human plasma
The purification of HRG from human plasma was performed as previously described[21]. Fresh citrated human blood was centrifuged at $2800 \times g$ for 10 min in the presence of 1 μg/mL prostaglandin E1. Plasma was collected, supplemented with cocktail inhibitors (leupeptin 10 μg/mL, aprotinin 10 μg/mL, NaF 50 mM, NaVO$_4$ 13 mM, pepstatin 10 μM, PMSF 1 mM) and centrifuged at $2800 \times g$ for 10 min again. After removing the residual cellular debris, plasma was centrifuged at $10,000 \times g$ at 4 °C for 30 min. The visible lipid layer on top was removed and imidazole was added to 5 mM. The sample was subjected to chromatography using a Ni-NTA agarose column (7Sea Biotech, China) pre-equilibrated with Tris-buffered saline (TBS: 20 mM Tris-HCl, 150 mM NaCl, pH 7.4). The column was washed in the following order with 10 volumes of each buffer: (1) TBS (pH 7.4) containing 5 mM imidazole, (2) TBS (pH 7.4) containing 5 mM imidazole and 10 mM ε-amino caproic acid, (3) TBS (pH 7.4) containing 80 mM imidazole and 10 KIU/mL aprotinin, and (4) TBS containing 100 mM imidazole and 10 KIU/mL aprotinin. HRG was eluted using TBS (pH 7.5) containing 250 mM imidazole, concentrated and buffer exchanged with PBS using a Millipore filter tube. The purity of HRG was determined to be >95% using Coommassie blue staining and immunoblotting analysis.

### Mechanism-based kinetic trapping of PDI substrates
The structure of PDI includes 2 catalytically active domains *a* and *a'* with each containing a Cys-Gly-His-Cys motif (Cys$_{53}$Gly$_{54}$His$_{55}$Cys$_{56}$ in *a* domain and Cys$_{397}$Gly$_{398}$His$_{399}$Cys$_{400}$ in *a'* domain, denoted as CCCC). Reduction of the substrate disulfides occurs through formation of a transient disulfide bond between the N-terminal Cys in PDI catalytic motif and the target Cys on substrate. Breakage of the mixed complexes requires the C-terminal Cys (Cys$_{56}$ in *a* domain and Cys$_{400}$ in *a'* domain) to attack the transient disulfide bond, therefore releasing the substrate in the reduced form. Mutation of the C-terminal Cys to Ala in the Cys-Gly-His-Cys motif in *a* domain (i.e., Cys$_{53}$Gly$_{54}$His$_{55}$Ala$_{56}$, denoted as CACC), in *a'* domain (i.e., Cys$_{397}$Gly$_{398}$His$_{399}$Ala$_{400}$, denoted as CCCA), or both domains (i.e., Cys$_{53}$Gly$_{54}$His$_{55}$Ala$_{56}$ and Cys$_{397}$Gly$_{398}$His$_{399}$Ala$_{400}$, denoted as CACA), results in the stabilization and accumulation of the transient complexes. Mutation of all the catalytic Cys to Ala (i.e., Ala$_{53}$Gly$_{54}$His$_{55}$Ala$_{56}$ and Ala$_{397}$Gly$_{398}$His$_{399}$Ala$_{400}$, denoted as AAAA) leads to the inert variant of PDI (Fig. 1a). These PDI variants were cloned into a pT7-FLAG-SBP vector (Sigma) to add an N-terminal Flag epitope for detection by immunoblotting and a C-terminal Streptavidin Binding Peptide for purification by chromatography.

Mechanism-based kinetic trapping and the validation of HRG as a redox substrate of PDI was performed as previously described[20]. In brief, PDI variants (PDI-CCCC, PDI-CACC, PDI-CCCA, PDI-CACA and PDI-AAAA) (500 μg) were pre-reduced with 20 mM dithiothreitol (DTT) at 0 °C for 20 min, desalted using Zeba spin column (Thermo), and added into recalcified (1 mM) platelet-rich plasma (5 mL) in the presence of 5 mM peptide Gly-Pro-Arg-Pro-NH$_2$ (GPRP) (Bachem). The clotting reaction was initiated with collagen (Chrono-Log) (10 μg/mL) and human α-thrombin (Sigma) (0.2 U/mL), incubated at 23 °C for 20 min, and terminated by the addition of 20 U/mL hirudin and 20 mM N-ethylmaleimide. The supernatant was collected and precipitated using streptavidin-agarose beads (Pierce). The PDI variants and their disulfide-linked complexes were eluted using 3 mM biotin and subjected to immunoblotting analysis under non-reducing and reducing conditions. PDI and HRG were simultaneously detected using mouse anti-FLAG tag (Cell Signaling Technology, 1:1000) on the PDI variants

and rabbit anti-human HRG (GeneTex, 1:2000), and then visualized using Alexa-488-conjugated goat anti-mouse IgG and Alexa-647-conjugated goat anti-rabbit IgG (Life Technologies), respectively. Dual immunofluorescence was imaged on an ImageQuant LAS4000 (GE Healthcare). Mechanism-based kinetic trapping with ERp57 variants was performed using the same method.

## Identification of PDI-cleaved disulfide bonds in HRG

Liquid chromatography, mass spectrometry (LC-MS) and data analysis were performed as previously described[26,60]. Recombinant PDI was reduced with 10 mM DTT at 25 °C for 30 min and desalted into PBS using Zeba spin column (Thermo Fisher). Reduced PDI was incubated with 2 μg of purified human HRG at 25 °C for 30 min. 2-iodo-N-phenylacetamide ([12]C-IPA) was added to the reactions to a final concentration of 5 mM followed by incubation at 25 °C for 1 h in dark. Proteins were analyzed on non-reducing SDS-PAGE and stained with Coomassie blue (BioRad). Bands corresponding to HRG were excised, destained with 25 mM sodium bicarbonate buffer containing 50% acetonitrile, dried in 100% acetonitrile followed by incubation with 40 mM DTT at 56 °C for 30 min. After repeated washing, the fully reduced proteins were alkylated with 5 mM 2-iodo-N-phenylacetamide where all 6 carbon atoms of the phenyl ring have a mass of 13 ([13]C-IPA, Cambridge Isotopes). The gel slices were washed, dried and deglycosylated with 5 Units of PNGase F (Sigma) at 37 °C overnight, followed by digestion with 12.5 ng/μL of chymotrypsin (Roche) in 25 mM NH$_4$HCO$_3$ containing 10 mM CaCl$_2$ at 25 °C overnight. Reactions were stopped by adding 5% (v/v) formic acid and peptides eluted twice from the gel slices with 30 μL of 5% (v/v) formic acid and 50% (v/v) acetonitrile. Peptides were dried, reconstituted in 1% (v/v) trifluoroacetic acid, and desalted using C18 ziptip (Millipore). Dried peptides were reconstituted in 12 μL of 0.1% formic acid, and then resolved (200 ng) on a 35 cm × 75 μm C18 reverse phase analytical column using a 2-35% acetonitrile gradient over 22 min with a flow rate of 250 nL/min (Thermo Fisher Ultimate 3000 HPLC). The peptides were ionized by electrospray ionization at +2.0 kV. Tandem mass spectrometry analysis was carried out on a Q-Exactive Plus mass spectrometer using higher energy collisional-induced dissociation fragmentation. The data-dependent acquisition method acquired MS/MS spectra of the top 10 most abundant ions with charged state ≥ 2–5 at any one point during the gradient. MS/MS spectra were searched against the Swissprot reference proteome using an external search engine Mascot (Version 2.7, Matrix Science) or HRG protein sequence (Uniprot identifiers P04196) using Byonic™ (Version 3.0, Protein Metrics). Precursor mass tolerance and fragment tolerance were set at 10 ppm and the precursor ion charge state to 2+, 3+ and 4+. Variable modifications were defined as oxidized Met, deamidated Asn/Gln, iodoacetanilide derivative Cys, and [13]C-iodoacetanilide derivative Cys with full chymotrypsin cleavage of up to 5 missed cleavages. The abundance of the different redox forms of the cysteines were calculated from the relative ion abundance of peptides labeled with [12]C-IPA and/or [13]C-IPA. The criteria for HRG cysteine-containing peptides analyzed to determine disulfide bond redox state is described in Supplementary Table S1. To calculate ion abundance of peptides, extracted ion chromatograms were generated using the XCalibur Qual Browser software (v2.1.0; Thermo Scientific). The area was calculated using the automated peak detection function built into the software.

The fraction of reduced disulfide bonds was measured from the relative ion abundance of peptides containing [12]C-IPA and [13]C-IPA. To calculate ion abundance of peptides, extracted ion chromatograms were generated using the XCalibur Qual Browser software (v2.1.0; Thermo Scientific). The area was calculated using the automated peak detection function built into the software. The ratio of [12]C-IPA and [13]C-IPA alkylation represents the fraction of the cysteine in the population that is in the reduced state.

For the analysis of disulfide pairing by mass spectrometry, purified human plasma HRG (3 μg) was alkylated with [12]C-IPA and resolved on non-reducing SDS-PAGE. Protein bands were excised, washed, dried and deglycosylated by PNGase F at 37 °C overnight. Gel pieces were washed and subjected to protease digestion as described above. LC-MS was performed as previously described[27]. Briefly, peptides (200 ng) were resolved on a 35 cm × 75 μm C18 reverse phase analytical column using a 2-35% acetonitrile gradient over 22 min at a flow rate of 300 nL/min, ionized by electrospray ionization at +2.0 kV and analyzed on a Q-Exactive Plus mass spectrometer (Thermo Fisher) using higher energy collisional-induced dissociation fragmentation. The data-dependent acquisition method acquired MS/MS spectra of the top 5 most abundant ions with charged state ≥ 2–8 at any one point during the gradient. Disulfide-linked peptides were searched against human HRG protein sequence using Byonic analysis software for disulfide cross-linked peptides. Precursor mass tolerance and fragment tolerance were set at 10 ppm and the precursor ion charge state to 2+ to 8+. Variable modifications were defined as oxidized Met, deamidated Asn/Gln, iodoacetanilide and derivative Cys with full chymotrypsin cleavage of up to 5 missed cleavages. The false discovery rate (FDR) was set at 0.01 and the P-value was computed using a method called two-dimensional FDR and a probability distribution with an exponential right-hand tail[61]. Relative abundance of peptides was calculated by normalizing their sum of intensities to that of the C6–C486 peptide.

## Biomembrane force probe (BFP) detection on cellular affinity

The detailed procedures for BFP were described previously[32,62–64]. In brief, purified human HRG and FXIIa (Enzyme Research) were covalently labeled with maleimide-PEG3500-SCM (JenKem, USA) in carbonate/bicarbonate buffer (pH 8.5) at room temperature. To functionalize the beads, 5 μL of MPTMS beads were incubated at room temperature overnight in phosphate buffer (pH 6.8) with HRG-maleimide and streptavidin-maleimide (SA-MAL, Sigma) to generate the Probe beads, or FXIIa-maleimide alone to generate the Target beads. To prepare the Target beads coated with heparin, 5 μL of MPTMS beads were incubated with 6250 IU/mL heparin sodium solution (Sigma) at 4 °C overnight. To ensure the coating densities were comparable between studies, the beads were coated with the same batch of proteins following the same protocol.

For the BFP adhesion frequency assay, the Target bead coated with FXIIa or heparin was driven by a micropipette to approach and impinge onto the Probe bead coated with HRG until the impingement force detected reached 20 pN. The two beads were then allowed to remain in contact for a defined contact time ($t_c$ = 0.1-5 s for adhesion frequency measurement) and then subjected to retraction. Based on the BFP force spectroscopy traces, an 'Adhesion' event was detected if the tensile force was observed upon retraction, otherwise the event was deemed as 'No adhesion'. The specificity of the interactions was confirmed using a negative control where the Target bead was immobilized with streptavidin alone. The number of adhesion events was enumerated to calculate the adhesion frequency ($P_a$) after 50 touch cycles for each Probe-Target pair. Three to five Probe-Target pairs were measured at each $t_c$. Since $P_a$ is dependent on $t_c$, the measured $P_a$ over 0.1-5 s was fitted into the following equation: $P_a(t_c) = 1 - \exp\{-eK_a[1 - \exp(-k_{off}t_c)]\}$[65] where $eK_a$ is the cellular affinity[33] between the coated proteins and $k_{off}$ is the 2D off-rate of the interactions.

## Assays for the activities of FXIIa, thrombin and FXa

To evaluate FXIIa activity, purified human plasma HRG (final 0.12 μM) was treated with vehicle, human PDI-CCCC or PDI-AAAA (final 14 μM) in the presence of 0.12 mM DTT at 37 °C for 5 min. The samples were diluted 1:5 using Tyrode's buffer (130 mM NaCl, 12 mM NaHCO$_3$, 5 mM D-glucose, 5 mM KCl, 10 mM Hepes, 0.4 mM Na$_2$HPO$_4$, 1 mM MgCl$_2$, pH

7.4) with or without 1 µM $Zn^{2+}$. The diluted HRG-PDI solution (70 µL) was mixed with 20 µL of FXIIa (final 10 nM) or Tyrode's buffer, and 10 µL of chromogenic substrate S2302 (Diapharma) (final 0.4 mM) in a microtiter plate. The chromogenic reaction was monitored on a plate reader (BioTek Elx800, USA) at 405 nm every 10 min for a total of 60 min.

In a separate experiment, human α-thrombin (final 5 nM) were mixed with gradient concentrations of purified human HRG (final 1, 3, 10, 30 nM). The activity of thrombin was detected by the cleavage of substrate Z-Gly-Gly-Arg-AMC·HCl (Bachem) (final 500 µM). The reaction in a total volume of 60 µL was monitored on a fluorescent plate reader (BMG Labtech, Germany) at 390 nm/460 nm (Ex/Em) every min for a total of 20 min. Alternatively, human FX (Enzyme Research) (final 125 nM) was mixed with FXa substrate Biophen CS-11 (Hyphen BioMed) (final 150 µM), TF (Innovin, Siemens) (final 1 pM) and gradient concentrations of purified human HRG (final 1, 3, 10, 30 nM) in the presence of 5 mM $CaCl_2$. The reaction in a total volume of 100 µL was initiated by the addition of 10 µL of human FVIIa (Enzyme Research) (final 0.6 nM). The activity of FXa was determined by measuring the OD at 405 nm every min for a total of 120 min.

### Immunofluorescence staining
Pooled normal human plasma was treated with human PDI-CCCC or PDI-AAAA (final 13 µM) in the presence of 0.3 mM DTT at 37 °C for 5 min. The reaction was diluted 1:10 using serum-free endothelial cell medium (ECM, ScienCell) containing 10 µM $Zn^{2+}$. The samples were added onto pre-washed HUVECs and incubated at 37 °C for 30 min. In a separate experiment, HUVECs were pre-treated with heparanase (Cloud-clone, China) at 100 or 300 ng/mL for 24 h before the addition of PDI-treated human plasma. The cells were then washed with PBS, fixed with paraformaldehyde, blocked with 10% goat serum, and stained with rabbit antibodies against HRG (BBI Life Sciences, 1:100) or antithrombin (Affinity Biosciences, 1:100) followed by CoralLite488-conjugated goat anti-rabbit IgG (Proteintech).

In a parallel experiment, pooled mouse plasma was treated with mouse PDI-CCCC or PDI-AAAA (final 13 µM) in the presence of 0.3 mM DTT at 37 °C for 5 min. The reaction was diluted 1:10 using DMEM containing 100 µM $Zn^{2+}$ and 1 µM argatroban (Aladdin, Shanghai). The samples were added onto pre-washed bEnd.3 cells and incubated at 37 °C for 30 min. The cells were then washed with PBS, fixed with paraformaldehyde, blocked with 10% goat serum, and stained with mouse antibody against HRG (Santa Cruz Biotech, 1:100) or rabbit antibody against antithrombin (Affinity Biosciences, 1:100) followed by CoralLite488-conjugated goat anti-mouse IgG (Proteintech) or CoralLite488-conjugated goat anti-rabbit IgG (Proteintech). The samples were imaged on a Zeiss LSM780 confocal microcopy.

### Solid-phase binding of HRG
Solid-phase binding of HRG was performed as previously described[66]. To determine the effect of PDI on HRG binding on heparin surface, purified human plasma HRG (final 1 µM) was treated with gradient concentrations of human PDI-CCCC or PDI-AAAA (final 0.1, 0.3, 1.0, 3.0, and 10.0 µM) or vehicle in the presence of 10 µM DTT at 37 °C for 10 min. The reaction was terminated with 20 mM N-ethylmaleimide, and diluted 1:100 with reaction buffer (50 mM Hepes, 150 mM NaCl, 0.1% Tween20, pH 7.4). The samples were then incubated on heparin-coated microtiter plates (Bioworld) in the presence or absence of 1 µM $Zn^{2+}$ at 37 °C for 2 h. To determine the effect of PDI on HRG binding on FXIIa, microtiter plates were coated with FXIIa (100 nM) at 4 °C overnight. Purified plasma HRG (final 0.35 µM) was treated with PDI-CCCC or PDI-AAAA (final 14 µM) or vehicle in the presence of 0.12 mM DTT at 37 °C for 10 min. The samples were diluted 1:7 and incubated on pre-coated microtiter plates with or without 1 µM $Zn^{2+}$ at 37 °C for 1 h. To determine the effect of cations on HRG binding, purified human HRG (10 nM) was incubated on microtiter plates pre-coated with heparin (625 U/mL) in the presence of gradient concentrations of $Ca^{2+}$ (0.3, 1, 3 mM) or $Zn^{2+}$ (1,

3, 10, 30 µM) for 2 h. In all experiments HRP-conjugated mouse-anti human HRG (Angio-Proteomie, 1:10000) was added into washed plates and incubated for 1 h. After washing the TMB substrate was added and the plate was measured at OD 450 nm to quantitate the bound HRG.

In a separate experiment, recombinant mouse HRG: WT- and gof-variant (1 nM for heparin or 30 nM for FXIIa) were incubated in the presence of $Zn^{2+}$ (1 µM) on microtiter plates pre-coated with heparin (625 U/mL) or FXIIa (100 nM) for 1 h. Goat anti-mouse HRG (R&D, 1:10000) was added after washing and incubated for 1 h. After removing the primary antibody, HRP-conjugated rabbit anti-goat (Proteintech) was added and incubated for 1 h. TMB substrate was used to quantitate the bound HRG by measuring the OD at 450 nm.

### Cell-based ELISA assay
Pooled normal human plasma was treated with PDI-CCCC or PDI-AAAA (final 20 µM) in the presence of 0.3 mM DTT at 37 °C for 5 min, and then diluted 1:10 using serum-free ECM. The treated samples were added onto HUVECs pre-seeded in a 96-well plate (100 µL per well) in the presence or absence of 10 µM $Zn^{2+}$ and incubated at 37 °C for 1 h. The plate was then washed with PBS, fixed with paraformaldehyde, and blocked with 3% BSA for 1 h. To examine the level of HRG binding on HUVECs, HRP-conjugated mouse-anti human HRG (Angio-Proteomie, 1:10000) was added for 1 h. Alternatively, to detect the level of antithrombin binding, rabbit anti-human antithrombin (Sangon Biotech, China, 1:2500) was added for 1 h followed by incubation with HRP-conjugated goat anti-rabbit IgG for 1 h. The bound antigen was quantified with the addition of TMB chromogenic substrate and examined on a microtiter plate reader.

### Thrombin generation assay
Thrombin generation assay (TGA) induced by TF on endothelial cells was performed using a method modified from previous protocols[43,67]. In brief, citrated plasma (30 µL) collected from adult mice (8-16 weeks old), including both males and females, was added into HUVECs seeded on a round-bottom microtiter plate, mixed with 5 µL of phospholipids (Rossix Phospholipid, diaPharma) (final 4 µM), 5 µL of recombinant TF (Innovin, Siemens) (1:8,000 dilution, final 0.75 pM), and 10 µL of diluted substrate Z-Gly-Gly-Arg-AMC·HCl (Bachem) (final 500 µM). The reaction was initiated by the addition of 10 µL of $CaCl_2$ (final 16.7 mM). Hydrolysis of the fluorogenic substrates was detected every 20 s for a total of 20 min using an automated Fluoroskan Ascent (Thermo Scientific) at 390 nm/460 nm (Ex/Em). The plasma sample from each mouse was measured in triplicate. The reaction slope at each time point was calculated and plotted over time. The peak height and AUC were additionally analyzed to characterize the amount of thrombin generation.

To analyze the influence of PDI on the inhibitory effect of HRG on FXIIa activity, FXIIa-induced TGA was performed using mouse plasma double deficient in HRG and FXII (DKO). Recombinant human PDI variants (CCCC and AAAA) or buffer only (vehicle group) were pre-reduced with 25 mM DTT on ice for 30 min and desalted into Tyrode's buffer using Zeba spin column (Thermo Fisher). Reduced PDI was incubated with purified human HRG with 8:1 molar ratio at 37 °C for 5 min. The PDI-HRG mixture (72 µL) was added into FXIIa (8 µL, final 120 nM) and incubated at 37 °C for 5 min. The reconstituted clotting system was composed of 30 µL of citrated DKO plasma, 5 µL of phospholipids (final 4 µM), 5 µL of PDI-HRG-FXIIa mixture (final PDI: 1.6 µM, HRG: 0.2 µM, FXIIa: 10 nM), and 10 µL of Z-Gly-Gly-Arg-AMC·HCl (Bachem) (final 500 µM) in a round-bottom microtiter plate. The reaction was initiated by the addition of 10 µL of $CaCl_2$ (final 16.7 mM). Hydrolysis of the fluorogenic substrates was detected every 20 s for a total of 20 min using an automated Fluoroskan Ascent (Thermo Scientific) at 390 nm/460 nm (Ex/Em). In a parallel experiment TGA was performed using aPTT reagent in $Hrg^{-/-}$ plasma supplemented with recombinant mouse HRG. Recombinant mouse PDI variants (12 µM) or vehicle were pre-reduced with DTT, desalted as described before, and then incubated

with recombinant mouse HRG (1.5 µM) at 37 °C for 5 min. The PDI-HRG mixture (20 µL) was added into $Hrg^{-/-}$ plasma (10 µL) followed by addition of aPTT reagent (6% v:v, 10 µL). After incubation at 37 °C for 10 min the fluorogenic substrate (3 mM, 10 µL) and $CaCl_2$ (6 mM, 10 µL) were added sequentially to initiate the clotting reaction. Hydrolysis of the substrates was measured as described before.

### Measurement of antithrombin activity

The activity of antithrombin was measured using a commercial assay kit (Stachrom, Stago) based on the protocol from the manufacturer. In brief, citrated mouse plasma (2.5 µL) was mixed with OWREN-KOLLER dilution buffer (47.5 µL) in a microtiter plate followed by addition of reconstituted thrombin solution (50 µL) with or without heparin. Alternatively, diluted plasma was mixed with thrombin solution on the surface of pre-seeded HUVECs. The mixture was incubated for 1 min followed by addition of thrombin substrate and measured for OD at 405 nm. The maximum thrombin activity ($OD_{max}$) was determined in a separate group containing only thrombin and substrates. The arbitrary antithrombin activity was defined as the difference in OD reflecting the thrombin cleavage of substrates using the equation (antithrombin = $OD_{max} - OD_{sample}$). All the data were normalized as percentage activity to the $WT$ group in the presence of heparin.

### Knockdown of plasma HRG using siRNA

This experiment utilized adult male mice (8-16 weeks old) from the C57BL/6 $F12^{-/-}$ strain. Control (4459405) and anti-HRG vivo-siRNA (s96987) were synthesized by Invitrogen and administered into $F12^{-/-}$ mice (180 µg of siRNA per 25 g mouse) through tail vein injection using Invivofectamine 3.0 Reagent (Invitrogen) according to the manufacturer's instructions. Laser-induced injury model of murine thrombosis was performed approximately 66 h after siRNA treatment. In the end of the experiment mouse blood was collected with acid-citrate-dextrose (ACD) solution by inferior vena cava (IVC). Equal amounts of plasma were analyzed by immunoblotting using anti-mouse HRG (R&D, 1:2000) and anti-mouse FXII (Cloud-Clone, China, 1:1500) for the level of plasma HRG and FXII, respectively, with anti-transferrin (Abclonal, China, 1:1000) as a loading control.

### FeCl3-induced thrombus formation in the carotid artery

Murine model of carotid artery thrombosis was performed as previously described[20,54]. This experiment utilized 8-9 adult mice (8-16 weeks old), including both males and females, from each strain C57BL/6 $WT$, $F12^{-/-}$, $Hrg^{-/-}$ and $DKO$. In brief, intravascular blood flow in the carotid artery was monitored for up to 90 min using a Doppler probe (Transonic TS420, UK) following $FeCl_3$-induced vessel damage. The time to full occlusion is defined as the first time point when complete flow cessation is achieved and sustains for additional 20 min. If the vessel remains open for 90 min, the occlusion time is defined as 90 min. The concentration of $FeCl_3$ (14%) was carefully titrated in a preliminary experiment so that most WT mice occluded within 10 min while approximately half of the $F12^{-/-}$ mice did not form occlusive thrombi during the observed time window. In some experiments, C57BL/6 mice were treated with vehicle or quercetin-3-rutinoside (Rutin) (Macklin, Shanghai) (25 µg/g body weight i.v.) before the application of $FeCl_3$ (12%). In the end of the experiment, the carotid artery was collected with both ends ligated to contain the intact thrombus. Cross sections were prepared from the paraformaldehyde-fixed paraffin-embedded samples and analyzed for the binding of HRG and antithrombin by immunohistochemistry using anti-HRG (Santa Cruz Biotech, 1:200) and anti-antithrombin (Affinity Biosciences, China, 1:200), respectively.

### Intravital microscopy of the laser-induced injury model of thrombosis

Intravital microscopy of the laser-induced injury model of thrombosis was performed as previously described[20,54,68]. This experiment examines the cremaster vasculature and thus utilizes only male mice (8-16 weeks old). Platelets and fibrin were visualized using Dylight-649-conjugated anti-CD42c antibody (Emfret Analytics, 0.1 µg/g body weight) and Alexa-488-conjugated 59D8 antibody (prepared in house, 0.5 µg/g body weight), respectively[54,68]. Anti-HRG (Angio-Proteomie) and anti-antithrombin (Affinity Biosciences) were labeled with Alexa-488 fluorophore (Invitrogen) and infused into WT mice by i.v. to visualized HRG (0.5 µg/g body weight) and antithrombin (0.5 µg/g body weight), respectively, during thrombus formation. In some experiments, eptifibatide (MedChemExpress) (10 µg/g body weight, repeated every 15-20 min), Rutin (5 µg/g body weight) or Bepristat 2a (15 µg/g body weight) was infused before vessel injury to block platelet accumulation or inhibit extracellular PDI activity, respectively. To compare the in vivo binding of different variants of mouse HRG, recombinant WT- and gof-mutants of mouse HRG were labeled with Alexa-488 fluorophore (Invitrogen), infused into $Hrg^{-/-}$ mice (50 µg/mouse), and visualized directly using the intravital microscopy. To compare the effects of different variants of mouse HRG on thrombus formation, recombinant WT- and gof-mutants of mouse HRG were infused into $Hrg^{-/-}$ mice (250 µg/mouse), followed by visualization of platelets and fibrin using Dylight-649-conguated anti-CD42c antibody and Alexa-488-conjugated 59D8 antibody, respectively. Images were analyzed using Slidebook v6.0 (Intelligent Imaging Innovations). The median fluorescence intensity over time were calculated and the AUC for each thrombus was quantified.

In vivo 3D scanning of the cremaster arterioles was acquired with a resonant scanning two-photon microscope (Scientifca) at 30 fps and a mode-locked Ti:sapphire laser (Mai Tai eHP DeepSee, Spectra-Physics) at 800 nm through a Nikon 16×/0.8 NA objective. HRG and CD31 were visualized in the growing thrombus using Alexa-488-conjugated anti-HRG (0.5 µg/g body weight) (Angio-Proteomie) and Alexa-647-conjugated anti-CD31 (0.5 µg/g body weight) (Biolegend), respectively. The green and red emission fluorescence was separated by a 565 nm dichroic mirror and detected by GaAsP PMTs. Image stacks ($512 \times 512$ pixel, $182.5 \times 182.5$ µm$^2$, 5-µm z-step size, 500-frame average/focal plane) of the cremaster arteriole were taken immediately following the laser-induced endothelial lesion at the focal plane by the same mode-locked Ti:sapphire laser at 800 nm (0.06 mW/µm$^2$ for 3 s). The colocalization of HRG and CD31 was analyzed with MATLAB2023 (MathWorks). Briefly, the pixel intensity of the green (HRG) and red (CD31) channels in the regions of interest (ROIs) was extracted and the subthreshold pixels were removed from further analysis. The Pearson's correlation coefficients for pixels with intensity above the threshold in each ROI was calculated.

### Data analysis

All measurements were taken from distinct samples. The representative results in the figures were from at least 3 independent experiments with similar results. All data were presented as Mean ± SEM and analyzed using GraphPad Prism 8.0. The statistical tests used in each figure were indicated in the respective legend. Two-tailed Welch's $t$-test was used to evaluate the difference between two groups of data. To compare three or more groups of data, Welch's ANOVA test was performed. In the laser-induced injury model of thrombosis two tailed Mann–Whitney $U$-test was used for binary comparison, and Kruskal–Wallis test with Dunn's multiple comparison was used to compare three or more groups. The exact $P$-value was indicated on the figures.

### Reporting summary

Further information on research design is available in the Nature Portfolio Reporting Summary linked to this article.

## Data availability

The authors declare that the data supporting the findings of this study are available within the paper and its supplementary information files.

The Mass Spec data generated in this study have been deposited to the ProteomeXchange Consortium via the PRIDE[69] partner repository with the dataset identifier PXD050718 [https://www.ebi.ac.uk/pride/]. The relevant raw data from each figure are provided in the Source Data file. Source data are provided with this paper.

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

## Acknowledgements

This work was supported in part by the National Natural Science Foundation of China (92169114 and 81802932) (C.F.), the Natural Science Fund for Distinguished Young Scholars of Hubei Province (2022CFA054) (C.F.), the National Key R&D Program of China (2018YFE0113600) (C.F.), the Global Research Award and the Supplement Award from the American Society of Hematology (C.F.), a HUST-QMUL joint research grant (2022-HUST-QMUL-SPRF-01) (C.F.), a HUST-USYD-JRC grant (C.F.), the Open Project of Hubei Key Laboratory of Wudang Local Chinese Medicine Research from the Hubei University of Medicine (WDCM2023001/2022001) (X.X.), a National Health and Medical Research Council (NHMRC) of Australia Ideas Grant (APP2003904) (L.A.J.), an NSW Cardiovascular Capacity Building Program Early-Mid Career Researcher Grant (L.A.J.), MRFF Cardiovascular Health Mission Grants (MRF2016165 and MRF2023977) (L.A.J.), and a National Heart Foundation Vanguard Grant (106979) (L.A.J.). L.A.J. is an Australian Heart Foundation Future Leader Fellow Level 2 (105863) and a Snow Medical Research Foundation Fellow (2022SF176).

## Author contributions

C.F. conceptualized and supervised the study and wrote the first draft. K.L., S.C., J.C., H.J.W., X.Y., S.R.B., H.W., Z.T., N.T., A.Y., S.Y. performed the experiments. K.L., S.C., X.X., J.C., Y.H., J.W., S.J., Y.W., A.H.S., L.A.J., P.J.H. and C.F. contributed to data analysis and edited the manuscript.

## Competing interests

S.R.B. is a current employee with Pfizer. All other authors declare they have no competing interests.

## Additional information

[1]Department of Pharmacology, School of Basic Medicine, Tongji Medical College and State Key Laboratory for Diagnosis and Treatment of Severe Zoonotic Infectious Diseases, Huazhong University of Science and Technology, Wuhan 430030 Hubei, China. [2]Department of Pharmacology, School of Basic Medicine, Guizhou University of Traditional Chinese Medicine, Guiyang 550025 Guizhou, China. [3]The Key Laboratory for Drug Target Researches and Pharmacodynamic Evaluation of Hubei Province, Wuhan 430030 Hubei, China. [4]Tongji-Rongcheng Center for Biomedicine, Huazhong University of Science and Technology, Wuhan 430030 Hubei, China. [5]The Centenary Institute, University of Sydney, Sydney, NSW 2006, Australia. [6]School of Biomedical Engineering, Faculty of Engineering, The University of Sydney, Darlington, NSW 2008, Australia. [7]The University of Sydney Nano Institute (Sydney Nano), The University of Sydney, Camperdown, NSW 2006, Australia. [8]Department of Neurobiology, School of Basic Medicine, Tongji Medical College, Huazhong University of Science and Technology, Wuhan 430030 Hubei, China. [9]Division of Hemostasis and Thrombosis, Beth Israel Deaconess Medical Center, Harvard Medical School, Boston, MA 02215, USA. [10]Department of Clinical Laboratory, Union Hospital, Tongji Medical College, Huazhong University of Science and Technology, Wuhan 430030 Hubei, China. [11]Department of Clinical Laboratory, Tongji Hospital, Tongji Medical College, Huazhong University of Science and Technology, Wuhan 430030 Hubei, China. [12]The Cyrus Tang Hematology Center, Soochow University, Suzhou 215123 Jiangsu, China. [13]Department of Vascular Surgery, Renji Hospital, School of Medicine, Shanghai Jiao Tong University, Shanghai 200127, China. [14]School of Stomatology, Tongji Medical Collage, Huazhong University of Science and Technology, and the Key Laboratory of Oral and Maxillofacial Development and Regeneration of Hubei Province, Wuhan 430030 Hubei, China. [15]Department of Endocrinology, Institute of Geriatric Medicine, Liyuan Hospital, Tongji Medical College, Huazhong University of Science and Technology, Wuhan 430030 Hubei, China. [16]Department of Medicine, Hematology, University Hospitals Cleveland Medical Center and Case Western Reserve University, Cleveland, OH 44106, USA. [17]These authors contributed equally: Keyu Lv, Shuai Chen, Xulin Xu. ✉e-mail: fangc@hust.edu.cn

