## [Peer Review File · Nature Communications]

Protein Disulfide Isomerase Cleaves Allosteric Disulfides in Histidine-Rich Glycoprotein to Regulate ThrombosisREVIEWER COMMENTS

Reviewer #1 (Remarks to the Author):

In this manuscript, Lv et al identify HRG as a substrate reduced by PDI and then investigate the complex mechanisms by which PDI regulates thrombosis in an HRG-dependent manner. The authors use a sophisticated and comprehensive set of in vitro and in vivo approaches to develop a model where PDI mediated reduction of HRG has competing procoagulant (neutralization of heparin sulfate) and anticoagulant (negative regulation of factor XIIa activity). Overall the studies are carefully designed and rigorously performed. Multiple complementary animal models are used. Limitations of the study emerge from the still incompletely understood mechanisms by which HRG regulates coagulation and the large number of substrates known to interact with both HRG and PDI, making this a very complicated problem, but this is a substantial advancement. The manuscript is generally very clear and should be of general interest to the readership. Several concerns could be addressed below to strengthen this interesting study:

Major concerns:

1. The major premise of the mechanism-based trapping strategy employed presumes PDI is reducing its substrate effectors. As such, most of the biochemical experiments use PDI that has been pre-reduced (apparently with dilution but not removal of DTT). The extracellular redox potential in plasma, as reflected by glutathione and cysteine buffer systems (PMID 1592968), is higher than PDI (PMID 8329391). Certainly extracellular proteins can be reduced, but the authors should discuss how this might apply in a physiologic system without external redox equivalents.
2. PDI has isomerase and other chaperone functions independent of redox catalysis that are not necessarily impaired by the AAAA mutant. Experiments in Fig. 3 compare AAAA and CCCC (+/- zinc), but do not include a vehicle, BSA, or similar control that enables one to distinguish the contribution of zinc alone from AAAA in these assays. Addition of such a control would be particularly helpful in the non-standard XIIa initiated thrombin generation assays, where the magnitude of the observed effects on thrombin generation is actually quite small. What happens if TGA is performed using a traditional contact activator (as

opposed to XIIa) to activate endogenous XII in plasma from single deficient Hrg^{-/-} mice spiked with PDI treated HRG? This would at least ensure more physiologic levels of factor XIIa in the system.

3. HRG binding at sites of injury is convincingly reduced. However, alternative explanations as to the mechanism are possible. Rutin reduces platelet and fibrin accumulation in the laser injury model. HRG is known to bind fibrinogen and remain bound upon conversion to fibrin (PMID 21757718), so is this a PDI independent mechanism that would enable HRG accumulation at sites of injury (perhaps explaining only partial reduction of HRG binding by eptifibatide)? The eptifibatide studies in Fig. 4a-e provide reassurance that HRG binding is not entirely platelet (and platelet zinc) dependent, but complicates interpretation of the rutin experiments in Fig. 5, where reduced platelet zinc would be expected and the authors have already shown that HRG interaction with FXII and HS are zinc dependent.

Minor suggestions:

-The first sentence of the abstract (line 65-6) suggesting PDI is the major difference between hemostasis and thrombosis is not well established and should be revised.

-line 117: condition should be plural

-line 133: would say "proposed" to act as a master switch, as this is unproven

-line 134: recommend alternative wording to "hemostatic proteins", PDI seems to be more important for thrombosis than hemostasis (references 18 -19)

-Some additional background information about PDI or description of the trapping mutants is likely required for most readers to understand, for example, what CACA means (perhaps include amino acid numbers)

-line 217: heparin isn't a target protein

-line 242: pretreated heparinase ("with" is omitted)

-line 300: word area should be removed

-line 423: recommend replacing "inappropriate" (who knows what is?) with "certain"

-line 430: delete "the trigger of"

-Fig. 1f and 8a appear to adapted from Poon et al (PMID 20971949), recommend differentiation of the figure and/or citation.

-Confirm that "medium" fluorescence intensity was intended throughout rather than mean

or median.

Reviewer #2 (Remarks to the Author):

The authors examined the involvement of protein disulfide isomerase (PDI) in the function of HRG in thrombosis. The authors identified human HRG as one of the substrates of human PDI which cleaves disulfide in the substrate molecule.

In the present manuscript, the authors identified three disulfide pairs in HRG and analyzed the functional roles of structural modifications of HRG in the initial phase and promotion phase of coagulation through the interaction with heparane sulfate (HS) and FXIIa, respectively.

I think that the aspect is interesting and unappreciated. The in vitro and in vivo methods used in the present study sound great. The results are reasonably described. However, I have several concerns on the manuscript listed below.

1. In the present experiments, HRG was purified from human plasma by one step affinity purification using Ni-NTA agarose column. Usually, it is very difficult to purify a plasma protein by one step procedure. There is no description about the purity of the HRG material used in the present study. The authors should provide the purity of their material.

2. The description on PDI mutants (CACC, CCCA, CACA, AAAA) is difficult to understand without referring to Bowley et al. (Ref. 20). Include a short explanation for the domain structure of PDI and the replacements of cysteine with alanine in a and a' domains in Methods and/or Results.

3. Figure 1e:

I do not understand the quantification data in Figure 1e. Were the data related to Fig 1d or not ?

4. Fig.3b:

What were the incubation conditions of cellular binding of HRG and antithrombin, incubation period and concentration ?

5. Figure 4e:

In vivo imaging were performed using Dylight-649-conjugated anti-CD42c (Red) and Alexa-488-conjugated anti-HRG (Green). Since Dylight-649-conjugated anti-CD42c was administered to three groups of mice and Alexa-488-conjugated anti-HRG (Green) was administered to the lower two groups. The labels in Figure 1a should clarify these procedures.

Why did the authors observe Alexa 488-non-specific signal on thrombus using control antibody ? Explain the significant accumulation of the control antibody.

6. The authors used different concentrations of Zn²⁺ (1, 3, 10, 30 microM) depending on the experiments. The authors should provide the reasons.

7. PDI inhibitor, isoquercetin, produces anti-thrombotic effect.

Although the authors speculated that PDI inhibitors may be another class of antithrombotic agents, the overall effects of PDI in allosteric modification of HRG contribute to antithrombotic effect. Thus, PDI inhibition may lead to the diminution of overall antithrombotic effect of HRG in relation to the present results. From this point, PDI inhibitors seem not to be antithrombotic agents straightly. Add the discussion on this issue.

8. The authors speculated that the replacement of antithrombin with allosterically modified HRG may lead to the loss of activity of antithrombin on the surface of endothelial cells. Thrombin formed on the surface of endothelial cells may be trapped by thrombomodulin and the released free antithrombin does not necessarily mean the loss of function. In the aqueous phase, antithrombin should produce anti-thrombin activity. Therefore, the initial procoagulant effect of HRG may come from another mechanism.

Minor

1. Line 265: "The" must be "the".

Responses to reviewers

Reviewer #1 (Remarks to the Author):

In this manuscript, Lv et al identify HRG as a substrate reduced by PDI and then investigate the complex mechanisms by which PDI regulates thrombosis in an HRG-dependent manner. The authors use a sophisticated and comprehensive set of in vitro and in vivo approaches to develop a model where PDI mediated reduction of HRG has competing procoagulant (neutralization of heparin sulfate) and anticoagulant (negative regulation of factor XIIa activity). Overall, the studies are carefully designed and rigorously performed. Multiple complementary animal models are used. Limitations of the study emerge from the still incompletely understood mechanisms by which HRG regulates coagulation and the large number of substrates known to interact with both HRG and PDI, making this a very complicated problem, but this is a substantial advancement. The manuscript is generally very clear and should be of general interest to the readership. Several concerns could be addressed below to strengthen this interesting study:

Major concerns:

1. The major premise of the mechanism-based trapping strategy employed presumes PDI is reducing its substrate effectors. As such, most of the biochemical experiments use PDI that has been pre-reduced (apparently with dilution but not removal of DTT). The extracellular redox potential in plasma, as reflected by glutathione and cysteine buffer systems (PMID 1592968), is higher than PDI (PMID 8329391). Certainly extracellular proteins can be reduced, but the authors should discuss how this might apply in a physiologic system without external redox equivalents.

Response: We appreciate the reviewer's comment. The essence of this question is how PDI is maintained as a reductase in the extracellular milieu, which is more oxidizing, without known reducing equivalents such as NADPH or NADH to fuel the reaction by regenerating its reductase activity. The plasma redox potential is reported to be -80 mV and -137 mV as reflected by the equilibrium of two buffer systems, Cys/CySS and GSH/GSSG, respectively (*Clin Sci* 2018, 132:1257). This value is higher than the reported redox potentials of PDI *a* and *a'* domains (-191 and -190 mV) (*Biochemistry* 1993, 32:6649). Therefore, reduction of disulfide bonds by PDI in extracellular proteins is feasible.

Currently, there are two proposed models that allow PDI reductase activity in the extracellular space (*J Biol Chem* 2019, 294:2949). In the first model PDI functions through a redox chain involving members of the vascular thiol isomerase family such as ERp72, ERp57, ERp5, etc., which work cooperatively by shuffling electrons among themselves and to substrate disulfides (*Blood* 2016, 128:893). This model is based on the evidence that these thiol isomerases can exchange electrons with each other (*Biochem J* 2015, 469:279). Comparison of their redox potentials suggests a

potential chain of electron flow from thioredoxin (-270 mV) to ERp72 (-217, -215, -220 mV) to ERp5 (-206, -211 mV) to PDI (-191, -190 mV) to ERp57 (-167, -156 mV), and to substrate disulfides (**J Biol Chem 2019, 294:2949**). However, this model may require other factors in the redox chain that is yet to be identified.

In the second model PDI functions through a single-turnover manner, i.e., consumed as a reductant secreted into the extracellular milieu to cleave one substrate without further reduction (**J Biol Chem 2019, 294:2949**). This single-turnover model is dependent on when and where PDI is secreted to provide precise control of disulfide cleavage in substrate and avoid the need for endogenous inhibitors which have not been identified to date. This model is supported by extracellular burst of PDI by activated platelets and endothelial cells during thrombosis (**J Clin Invest 2008, 118:1123; Blood 2010, 116:4665**).

Although in both scenarios the original sources that regenerate electrons to fuel the reactions on the cell surface are unknown, several mechanisms have been reported to control the redox state of thiol oxidoreductases/isomerases. For instance, in the cytoplasm thioredoxin is reduced by thioredoxin reductase using electrons from NADPH (**J Biol Chem 1989, 264:13963**). On platelet surface PDI is oxidized by Ero1 α (**J Biol Chem 2010, 285:29874; Redox Biol 2022, 50:102244**). In spite of the lack of reported mechanisms by which the reductase activity of extracellular PDI is regenerated, if it functions through the latter model, i.e. as a single-turnover reductant, it is more likely that PDI is pre-reduced by thioredoxin and/or NADPH in a segregated intracellular compartment before it is secreted to the cell surface (**Br J Pharmacol 2021, 178:2911; J Biol Chem 1989, 264:13963; J Biol Chem 2019, 294:2949**).

The Discussion has been updated to reflect this more detailed information in the revised manuscript.

2. PDI has isomerase and other chaperone functions independent of redox catalysis that are not necessarily impaired by the AAAA mutant. Experiments in Fig. 3 compare AAAA and CCCC (+/- zinc), but do not include a vehicle, BSA, or similar control that enables one to distinguish the contribution of zinc alone from AAAA in these assays. Addition of such a control would be particularly helpful in the non-standard XIIa initiated thrombin generation assays, where the magnitude of the observed effects on thrombin generation is actually quite small. What happens if TGA is performed using a traditional contact activator (as opposed to XIIa) to activate endogenous XII in plasma from single deficient Hrg-/- mice spiked with PDI treated HRG? This would at least ensure more physiologic levels of factor XIIa in the system.

Response: We appreciate this concern by the reviewer. Indeed, PDI has chaperon and isomerase activities which do not involve the net change of disulfide bonds (**Br J Pharmacol 2021, 178:2911**). However, it is unlikely that these redox catalysis-

independent functions of PDI have a major impact on HRG.

Firstly, it should be noted that BSA is not an appropriate control in this case because BSA itself binds to Zn^{2+} and consumes the ambient Zn^{2+} concentration in the reaction. Thus, we chose to use buffer only (vehicle) as the control. Comparison of the effect of vehicle with that of PDI-AAAA in the presence or absence of Zn^{2+} allows for distinguishing the contribution from PDI-AAAA.

Secondly, the current data suggest that the effects of PDI on HRG depend on its catalytic motifs. In the BFP experiment (**Fig. 2**), Zn^{2+} alone significantly increased the binding affinity of HRG to heparin and FXIIa. In the presence of Zn^{2+} , there is no significant difference between buffer alone (vehicle) and PDI-AAAA-treated group, whereas PDI-CCCC significantly increased the binding of HRG to heparin and FXIIa. Similarly, in the solid-phase binding assay (**Fig. 3a**), Zn^{2+} alone is sufficient to support the binding of HRG to heparin, albeit at basal level. PDI-AAAA showed no effect compared to vehicle (i.e. 0 μ M PDI) while PDI-CCCC increased the binding in a dose-dependent manner.

Thirdly, as the reviewer suggested, we added the buffer only (vehicle) group in the solid-phase binding assay for FXIIa (updated **Fig. 3e**), the chromogenic assay for FXIIa activity (updated **Fig. 3f**), and the thrombin generation assay induced by FXIIa (updated **Fig. 3g**). There is no significant difference in HRG binding to FXIIa between vehicle-treated and PDI-AAAA-treated groups in the presence or absence of Zn^{2+} (updated **Fig. 3e**). Further, there is no difference in FXIIa-induced thrombin generation between the reactions supplemented with vehicle-treated HRG and PDI-AAAA-treated HRG (updated **Fig. 3g**). In both assays, PDI-CCCC significantly enhanced the effects of HRG compared to PDI-AAAA. Note, when using the chromogenic assay, PDI-AAAA-treated HRG led to enhanced cleavage of pseudo-substrate by FXIIa compared to vehicle-treated HRG (updated **Fig. 3f**). This effect is attributed to the redox-independent chaperon activity of inert-PDI which possibly provides a scaffold to promote the recognition of the small-molecule pseudo-substrate by FXIIa. The same effect was also observed with ERp57 (data not shown). However, under physiological scenarios (e.g. thrombin generation in the plasma as shown in **Figs. 3g** and **3j**), PDI-AAAA did not influence the effect of HRG on FXIIa when compared to vehicle treatment.

Finally, as the reviewer suggested, we performed thrombin generation assay using a traditional contact activator (aPTT reagent from Stago) based on published protocols (**Blood 2015, 125:710; Front Cardiovasc Med 2022, 9:1008410**). In this experiment recombinant mouse HRG was pre-treated with vehicle, mouse PDI-AAAA or mouse PDI-CCCC, and then added into *Hrg*^{-/-} plasma (**Fig. 3j**). Consistent with the previous observation, addition of HRG significantly inhibited aPTT-induced thrombin generation in *Hrg*^{-/-} plasma. There was no significant difference between vehicle-treated and PDI-AAAA-treated HRG, whereas addition of PDI-CCCC-treated HRG further inhibited thrombin generation compared to PDI-AAAA-treated group, as reflected by prolonged lag time and peak time (**Fig. 3k**) and decreased peak height

and area under the curve (**Fig. 3I**).

Taken together, these results showed that the effects of PDI on HRG during coagulation is dependent on its CGHC motifs through redox catalysis.

3. HRG binding at sites of injury is convincingly reduced. However, alternative explanations as to the mechanism are possible. Rutin reduces platelet and fibrin accumulation in the laser injury model. HRG is known to bind fibrinogen and remain bound upon conversion to fibrin (PMID 21757718), so is this a PDI independent mechanism that would enable HRG accumulation at sites of injury (perhaps explaining only partial reduction of HRG binding by eptifibatide)? The eptifibatide studies in Fig. 4a-e provide reassurance that HRG binding is not entirely platelet (and platelet zinc) dependent, but complicates interpretation of the rutin experiments in Fig. 5, where reduced platelet zinc would be expected and the authors have already shown that HRG interaction with FXII and HS are zinc dependent.

Response: We thank the reviewer for the opportunity to explain this complex situation.

As the reviewer pointed out, HRG binds to fibrinogen and fibrin in a Zn^{2+} -dependent manner (*J Biol Chem* 2011, 286:30314). However, it is unlikely that fibrinogen- or fibrin-derived HRG had a major contribution to the total accumulation of HRG during thrombus formation.

Firstly, although HRG is present in platelets and released upon platelet activation (*Blood* 1983, 62:1016), the confocal intravital experiment using scanning two-photon microscopy showed that HRG accumulates mainly at the interface between platelet thrombi and endothelium (**Figs. 4f, 4g and 4h**). This location is consistent with the architectural characteristics of a growing thrombus in the laser-induced injury model where granule exocytosis, thrombin generation and fibrin deposition are restricted to the core region. Thus, the reduction of HRG accumulation in the presence of eptifibatide is not a consequence of elimination of platelet-derived HRG, but more likely due to reduced availability of platelet-derived Zn^{2+} in the local milieu. Secondly, there is a potential overlap in the binding sites on fibrinogen γ -chain for HRG and integrin (*J Biol Chem* 2011, 286:30314; *PLoS One* 2012, 77:e40033; *Matrix Biol* 2015, 48:89). The steric hindrance limits the ability of fibrinogen to serve a scaffold bridging HRG and integrin during thrombus formation. Thirdly, in the laser-induced injury model of thrombosis treatment with eptifibatide eliminates platelet accumulation without significantly affecting fibrin generation (*Proc Natl Acad Sci* 2007, 104:288; *Blood* 2010, 116:4665; *Blood* 2012, 120:647; *Pharmacol Res* 2021, 167:105540). Hence, the unaltered fibrin generation but decreased HRG accumulation by eptifibatide excludes a major role of fibrin in HRG accumulation during thrombosis. Taken together, fibrinogen- or fibrin-bound HRG, if any *in vivo* during thrombus formation, does not account for the majority of HRG accumulation

that is independent of PDI regulation.

In fact, in the laser-induced thrombosis model eptifibatide blocks integrin $\alpha\text{IIb}\beta\text{3}$ and thus platelet-platelet interactions during the propagation phase without affecting the initial single layer of integrin-independent platelet adhesion at the core region following vessel injury (*Blood* 2001, 98:1055; *Blood* 2012, 120:647). The current study showed that Zn^{2+} and PDI function in a synergistical manner with Zn^{2+} alone sufficient to support HRG binding to endothelium, albeit at basal level, whereas PDI markedly potentiated the binding (**Fig. 2, Figs. 3a and 3c**). Therefore, when treated with eptifibatide, the ambient Zn^{2+} secreted by the remaining layer of platelet adhesion and possibly PDI released by activated endothelial cells at the local milieu allow HRG binding at the basal level, as evidenced by the partial reduction of HRG accumulation in the presence of eptifibatide. Comparison of the fluorescent signal of HRG and platelets confirmed that the kinetics of HRG accumulation coincide with platelet adhesion during the initial phase of thrombus formation in the presence or absence of eptifibatide (**Figure I**). However, when treated with Rutin, a PDI inhibitor (*J Clin Invest* 2012, 122:2104) to block platelet accumulation (and therefore the burst of secreted Zn^{2+}) and fibrin generation as well as extracellular PDI secreted by injured endothelial cells, further reduction of HRG accumulation was observed (**Figs. 5c-5e**). This result was further confirmed using a synthetic small-molecule PDI inhibitor Bepristat 2a (*Nat Commun* 2016, 7:12579), which markedly inhibited *in vivo* platelet aggregation and HRG accumulation to a similar extent as Rutin (**Fig. S7**).

Taken together, the partial reduction of HRG accumulation in the presence of eptifibatide does not compromise our conclusion that the HRG binding mediated by PDI is dependent on Zn^{2+} . The Discussion has been updated to reflect this assessment in the revised manuscript.

Figure I. Comparison of the kinetics of HRG binding and platelet accumulation in the absence (a) or presence (b) of eptifibatide in the laser-induced thrombosis model. The grey area marked with an arrow indicates the initial phase of thrombus formation.

Minor suggestions:

- The first sentence of the abstract (line 65-6) suggesting PDI is the major difference between hemostasis and thrombosis is not well established and should

be revised.

Response: We agree with the reviewer and have revised this sentence as the following “The essence of difference between hemostasis and thrombosis is that the clotting reaction is a highly fine-tuned process. Vascular protein disulfide isomerase (PDI) represents a critical mechanism regulating the functions of hemostatic proteins.”

- line 117: condition should be plural

Response: We apologize for the error and have corrected it in the revised manuscript.

- line 133: would say “proposed” to act as a master switch, as this is unproven

Response: We fully agree with the reviewer and have modified this sentence as suggested.

- line 134: recommend alternative wording to “hemostatic proteins”, PDI seems to be more important for thrombosis than hemostasis (references 18 -19)

Response: We agree with the suggestion and have modified this sentence in the revised manuscript. The word “hemostatic proteins” has been changed to “thrombosis-related proteins”.

- Some additional background information about PDI or description of the trapping mutants is likely required for most readers to understand, for example, what CACA means (perhaps include amino acid numbers)

Response: We apologize for the lack of description on PDI mutants. The structure of PDI includes 2 catalytically active domains *a* and *a'* with each containing a Cys-Gly-His-Cys motif (Cys₅₃Gly₅₄His₅₅Cys₅₆ in *a* domain and Cys₃₉₇Gly₃₉₈His₃₉₉Cys₄₀₀ in *a'* domain, denoted as CCCC). Reduction of the substrate disulfides occurs through formation of a transient disulfide bond between the N-terminal Cys in PDI catalytic motif and the target Cys on substrate. Breakage of the mixed complexes requires the C-terminal Cys (Cys₅₆ in *a* domain and Cys₄₀₀ in *a'* domain) to attack the transient disulfide bond, therefore releasing the substrate in the reduced form. Mutation of the C-terminal Cys to Ala in the Cys-Gly-His-Cys motif in *a* domain (i.e., Cys₅₃Gly₅₄His₅₅Ala₅₆, denoted as CACC), in *a'* domain (i.e., Cys₃₉₇Gly₃₉₈His₃₉₉Ala₄₀₀, denoted as CCCA), or both domains (i.e., Cys₅₃Gly₅₄His₅₅Ala₅₆ and Cys₃₉₇Gly₃₉₈His₃₉₉Ala₄₀₀, denoted as CACA), results in the stabilization and accumulation of the transient complexes. Mutation of all the catalytic Cys to Ala (i.e., Ala₅₃Gly₅₄His₅₅Ala₅₆ and Ala₃₉₇Gly₃₉₈His₃₉₉Ala₄₀₀, denoted as AAAA) leads to the inert

variant of PDI.

A schematic depicting these variants has been added to Fig. S1 in the revised manuscript. The Methods and Results have also been updated accordingly.

- line 217: heparin isn't a target protein.

Response: We have changed the word "target proteins" to "target ligands" in the revised manuscript.

- line 242: pretreated heparinase ("with" is omitted)

Response: We apologize for the typo and have corrected it in the revised manuscript.

- line 300: word area should be removed

Response: We apologize for the error and have corrected it.

- line 423: recommend replacing "inappropriate" (who knows what is?) with "certain"

Response: We fully agree with the suggestion and have modified this sentence as suggested.

- line 430: delete "the trigger of"

Response: This sentence has been modified as suggested.

- Fig. 1f and 8a appear to be adapted from Poon et al (PMID 20971949), recommend differentiation of the figure and/or citation.

Response: We thank the reviewer for this comment. As suggested, updated schematics depicting the domain structures of human and mouse HRG have been added into the revised manuscript.

- Confirm that "medium" fluorescence intensity was intended throughout rather than mean or median.

Response: We apologize for the typo. It should be "median fluorescence intensity". The error has been corrected throughout the manuscript.

Reviewer #2 (Remarks to the Author):

The authors examined the involvement of protein disulfide isomerase (PDI) in the function of HRG in thrombosis. The authors identified human HRG as one of the substrates of human PDI which cleaves disulfide in the substrate molecule.

In the present manuscript, the authors identified three disulfide pairs in HRG and analyzed the functional roles of structural modifications of HRG in the initial phase and promotion phase of coagulation through the interaction with heparane sulfate (HS) and FXIIa, respectively.

I think that the aspect is interesting and unappreciated. The in vitro and in vivo methods used in the present study sound great. The results are reasonably described. However, I have several concerns on the manuscript listed below.

1. In the present experiments, HRG was purified from human plasma by one step affinity purification using Ni-NTA agarose column. Usually, it is very difficult to purify a plasma protein by one step procedure. There is no description about the purity of the HRG material used in the present study. The authors should provide the purity of their material.

Response: We appreciate this reviewer's request for this additional information. In this study HRG was purified from human plasma using Ni-NTA agarose column based on a previously published protocol (*Blood* 2011, 117:4134). Instead of one-step chromatography, the protocol consists of multiple steps of washing with increasing concentrations of imidazole in the following order: (1) TBS (pH 7.4) containing 5 mM imidazole, (2) TBS (pH 7.4) containing 5 mM imidazole and 10 mM ϵ -amino caproic acid, (3) TBS (pH 7.4) containing 80 mM imidazole and 10 KIU/mL aprotinin, and (4) TBS containing 100 mM imidazole and 10 KIU/mL aprotinin. HRG was eluted using TBS (pH 7.5) containing 250 mM imidazole. We verified the purity of the product by Commassie blue staining and immunoblotting analysis (**Figure II**). The results showed > 95% purity of HRG. We have added the purity information of HRG into the supplemental method describing the procedure at appropriate place.

Figure II. The purify of human HRG was verified by Commassie blue staining (a) and immunoblotting analysis (b) using variable amounts of proteins as indicated.

2. The description on PDI mutants (CACC, CCCA, CACA, AAAA) is difficult to understand without referring to Bowley et al. (Ref. 20). Include a short explanation for the domain structure of PDI and the replacements of cysteine with alanine in a and a' domains in Methods and/or Results.

Response: We apologize for the lack of clarity of the description on PDI mutants. The structure of PDI includes 2 catalytically active domains *a* and *a'* with each containing a Cys-Gly-His-Cys motif (Cys₅₃Gly₅₄His₅₅Cys₅₆ in *a* domain and Cys₃₉₇Gly₃₉₈His₃₉₉Cys₄₀₀ in *a'* domain, denoted as CCCC). Reduction of the substrate disulfides occurs through formation of a transient disulfide bond between the N-terminal Cys in PDI catalytic motif and the target Cys on substrate. Breakage of the mixed complexes requires the C-terminal Cys (Cys₅₆ in *a* domain and Cys₄₀₀ in *a'* domain) to attack the transient disulfide bond, therefore releasing the substrate in the reduced form. Mutation of the C-terminal Cys to Ala in the Cys-Gly-His-Cys motif in *a* domain (i.e., Cys₅₃Gly₅₄His₅₅Ala₅₆, denoted as CACC), in *a'* domain (i.e., Cys₃₉₇Gly₃₉₈His₃₉₉Ala₄₀₀, denoted as CCCA), or both domains (i.e., Cys₅₃Gly₅₄His₅₅Ala₅₆ and Cys₃₉₇Gly₃₉₈His₃₉₉Ala₄₀₀, denoted as CACA), results in the stabilization and accumulation of the transient complexes. Mutation of all the catalytic Cys to Ala (i.e., Ala₅₃Gly₅₄His₅₅Ala₅₆ and Ala₃₉₇Gly₃₉₈His₃₉₉Ala₄₀₀, denoted as AAAA) leads to the inert variant of PDI. As suggested, a schematic depicting these variants has been added to Fig. S1 in the revised manuscript. The Methods and Results have also been updated accordingly.

3. Figure 1e: I do not understand the quantification data in Figure 1e. Were the data related to Fig 1d or not?

Response: We apologize for the lack of clarification on this figure.

Human HRG contains 16 cysteines predicted to form 6 disulfide bonds based on the crystal structure of the N2 domain in rabbit HRG (*Blood* **2014**, **123:1948**). However, based on our differential cysteine alkylation, 8 rather than 6 disulfide bonds were predicted in human HRG. To confirm how the cysteines paired with each other as in Fig. 1d, we analyzed disulfide-linked peptides derived from HRG purified from human plasma following protease digestion without disulfide reduction by DTT. The identity of the label-free disulfide-linked peptides were directly analyzed by mass spectrometry using Byonic software (**Table S2**). As shown in Fig. 1e, the intensity of disulfide-linked peptides was quantified by Qual Browser and was expressed as a ratio of the respective peptide to C6-C486-containing peptide.

Depending on the composition of peptides and their affinity to the C18 column used in HPLC, peptides containing different disulfide bonds show dissimilar abundance to the peptide containing C6-C486. For instance, a ratio greater than 1 for C306-C309 indicates more peptide containing this bond bound to the C18 column as compared to the peptide containing C6-C486, whereas a ratio less than 1 for C390-C434 indicates less peptide containing this bond bound to the C18 column. It should

be noted that the relative abundance shown in Fig. 1e does not correlate with the redox state of cysteines determined in Fig. 1d since this label-free approach does not discriminate free and disulfide form of cysteines. For example, the C60-C71 and C87-C108 bonds could not be detected due to their crosslinking into tripeptide which has a molecular mass exceeding the limit of detection by the mass spectrometer. Nevertheless, this does not affect the quantification of their redox state since peptides containing C60-C71 and C87-C108 were subjected to reduction by DTT and cysteine alkylation in Fig. 1d. Hence, the purpose of Fig. 1e is only to confirm the pairing of cysteines in HRG and the quantification itself is not related to Fig. 1d. Fig. 1e also independently and experimentally confirmed that three disulfide bonds targeted by PDI, C306-C309, C390-C434, and C409-C410, exist in HRG.

We have included additional explanation in the figure legend (Fig. 1e) in the revised manuscript to explain the interpretations of the disulfide-linked peptide analysis.

4. Fig.3b: What were the incubation conditions of cellular binding of HRG and antithrombin, incubation period and concentration?

Response: We re-examined the data and experimental details for Fig.3b. The procedure was provided in supplemental methods under the section of “immunofluorescence staining” in the original manuscript. To mimic the physiological condition, cellular binding of HRG and antithrombin was performed using citrated plasma instead of purified proteins based on a protocol in our previous study (*Nat Commun* 2017, 8:14151). In brief, pooled normal human plasma was treated with human PDI-CCCC or PDI-AAAA (final 13 μ M) in the presence of 0.3 mM DTT at 37°C for 5 min. The reaction was diluted 1:10 using serum-free endothelial cell medium (ECM, ScienCell) containing 10 μ M Zn²⁺. The samples were added onto pre-washed HUVECs and incubated at 37°C for 30 min. The cells were then washed with PBS, fixed with paraformaldehyde, blocked with 10% goat serum, and stained with rabbit antibodies against HRG (BBI Life Sciences) or antithrombin (Affinity Biosciences) followed by CoralLite488-conjugated goat anti-rabbit IgG (Proteintech).

5. Figure 4e: In vivo imaging was performed using Dylight-649-conjugated anti-CD42c (Red) and Alexa-488-conjugated anti-HRG (Green). Since Dylight-649-conjugated anti-CD42c was administered to three groups of mice and Alexa-488-conjugated anti-HRG (Green) was administered to the lower two groups. The labels in Figure 1a should clarify these procedures. Why did the authors observe Alexa 488-non-specific signal on thrombus using control antibody? Explain the significant accumulation of the control antibody.

Response: We apologize for any confusion caused by inaccurate labeling in this figure. As the reviewer pointed out, the lower two groups were both administered with Alexa-488-conjugated anti-HRG with the bottom one additionally treated with

eptifibatide to eliminate platelets. The top group was infused with Alexa-488-conjugated isotype control for anti-HRG. All three groups were treated with Dylight-649-conjugated anti-CD42c. The labeling has been updated in this panel to reflect the two channels (green and red) used to visualize HRG and platelets, respectively.

Like other imaging techniques, the signal from the top group, which was infused with Alexa-488-conjugated IgG, served as the background to validate the significance of the signal from Alexa-488-conjugated anti-HRG. As shown in the quantification in Figs. 4b and 4e, on the log scale the background signal in the IgG group was minimal whereas the signal from anti-HRG-treated group was significantly higher, suggesting considerable amount of HRG accumulation at the site of injury. The involvement of IgG control as a background is an established approach in the laser injury-induced thrombosis model to investigate the accumulation of thrombus-residing proteins such as vitronectin (*Nat Commun* 2017, 8:14151), PDI (*J Clin Invest* 2008, 118:1123), ERO1 α (*Circ Res* 2023, 132:e206) and tissue factor (*Nat Med* 2002, 8:1175). To better reflect the quantification data, an updated set of representative images has been provided for the IgG control group.

6. The authors used different concentrations of Zn²⁺ (1, 3, 10, 30 microM) depending on the experiments. The authors should provide the reasons.

Response: We appreciate the reviewer's question.

Free Zn²⁺ binds to HRG with a K_d value of 225 nM and saturated binding was achieved at > 5 μ M (*Blood* 2011, 117:4134). Further, the effect of Zn²⁺ on HRG binding to FXIIa reached maximum at ~ 2.5 μ M Zn²⁺. Therefore, in the solid-phase binding assay (Figs. 3a and 3e) and FXIIa chromogenic assay (Fig. 3f) which utilized purified proteins in a reconstituted system, Zn²⁺ was added at 1 μ M, which is a subthreshold concentration to avoid any saturating effect and allow the detection of the effects of PDI.

In the immunofluorescence staining (Fig. 3b) and cell-based ELISA (Figs. 3c and 3d) which utilized human plasma in an *ex vivo* system, Zn²⁺ was used at 10 μ M. Previous study (*Blood* 2002, 99:3589) from one of the coauthors showed that activation of physiological concentration (e.g., 2.2×10^8 /mL) of platelets release Zn²⁺ to an ambient concentration of ~10 μ M. This concentration coincides with the plasma concentration of Zn²⁺ (10 ~ 20 μ M) (*J Clin Pathol* 1984, 37:1050).

Lastly, to examine the effect of different cations including Zn²⁺ and Ca²⁺ on HRG binding to heparin (Fig. S4), gradient concentrations of Zn²⁺ (1, 3, 10, 30 μ M) or Ca²⁺ (0.3, 1, 3 mM) were added into the *in vitro* system containing purified human HRG. Using gradient concentrations allows for the detection of the effect in a broader range.

7. PDI inhibitor, isoquercetin, produces anti-thrombotic effect. Although the authors

speculated that PDI inhibitors may be another class of antithrombotic agents, the overall effects of PDI in allosteric modification of HRG contribute to antithrombotic effect. Thus, PDI inhibition may lead to the diminution of overall antithrombotic effect of HRG in relation to the present results. From this point, PDI inhibitors seem not to be antithrombotic agents straightly. Add the discussion on this issue.

Response: This comment shows insight into this field. The fact that the absence of HRG enhanced thrombosis suggests that the role of FXIIa and the contact pathway is important for thrombosis and supersedes the effect of HRG on HS neutralization. Thus, as the reviewer pointed out, PDI inhibition may result in diminution of the overall antithrombotic effect of HRG. However, it should be noted that PDI, as a “master switch” (*Blood* **2016**, **128:893**), also acts on other substrates in addition to HRG. For example, we previously showed that PDI supports thrombosis by promoting the incorporation of vitronectin into a growing thrombus (*Nat Commun* **2017**, **8:14151**; *Br J Pharmacol* **2021**, **178:2911**). PDI also regulates the binding activities of platelet adhesive receptors including integrin $\alpha\text{IIb}\beta\text{3}$ (*Antioxid Redox Signal* **2016**, **24:16**), GPIIb α (*Circulation* **2019**, **139:1300**), etc. Inhibition of PDI led to attenuated vitronectin accumulation, reduced platelet aggregation and diminished fibrin generation (*Nat Commun* **2017**, **8:14151**). Therefore, the overall effect of PDI inhibitors, depending on the spectrum of substrates, is antithrombotic. Nevertheless, given the anticoagulant effect of HRG, suppression of HRG by PDI inhibitors provides a procoagulant potential which puts a “brake” on the potent antithrombotic effects caused by PDI inhibition. Thus, the PDI-HRG pathway may provide mechanistic insights for the clinical effects of PDI inhibitors such as Rutin and Isoquercetin, which inhibit thrombosis without causing significant bleeding (*J Clin Invest* **2012**, **122:2104**; *JCI Insight* **2019**, **4:e125851**). This section of Discussion has been updated in the revised manuscript to include these seemingly paradoxical observations into the sum of PDI and its inhibitors’ activities.

8. The authors speculated that the replacement of antithrombin with allosterically modified HRG may lead to the loss of activity of antithrombin on the surface of endothelial cells. Thrombin formed on the surface of endothelial cells may be trapped by thrombomodulin and the released free antithrombin does not necessarily mean the loss of function. In the aqueous phase, antithrombin should produce anti-thrombin activity. Therefore, the initial procoagulant effect of HRG may come from another mechanism.

Response: We appreciate the reviewer for this comment.

We evaluated the antithrombin activity in the plasma from *WT* and *Hrg*^{-/-} mice utilizing a commercial assay kit which measures the cleavage of chromogenic substrates by residual activity of excessive thrombin in the presence of fixed amount of heparin. Consistent with a previous report (*J Thromb Haemost* **2005**, **3:865**), *Hrg*^{-/-} plasma, when added with heparin, exhibited increased antithrombin activity compared to *WT* (**Fig. 6a** in the revised manuscript). However, there was no

significant antithrombin activity detected in the absence of heparin, suggesting the anticoagulant activity of antithrombin in the aqueous phase in the plasma *per se* is very slow (*J Clin Invest* 1984, 74:341). Given the fast kinetics during the initiation of arterial thrombosis, it is unlikely that the initial procoagulant effect of HRG is attributed to the altered antithrombin activity, if any, in the aqueous phase. In fact, on the surface of endothelial cells, HRG deficiency resulted in significantly enhanced antithrombin activity (**Fig. 6a** in the revised manuscript), confirming that the initial procoagulant effect of HRG is most likely mediated by inhibition of endothelial antithrombin.

Minor

- Line 265: “The” must be “the”.

We apologize for the typo. The error has been corrected in the revised manuscript.

REVIEWERS' COMMENTS

Reviewer #1 (Remarks to the Author):

All of my comments and concerns have been satisfactorily addressed.

Reviewer #2 (Remarks to the Author):

The authors addressed to all comments I provided and the replies to my questions are satisfactory. I think that the revised manuscript has been improved accordingly.